# Learn and Ensemble Bridge Adapters for Multi-domain Task Incremental Learning

**Ziqi Gu**[1], **Chunyan Xu**[1,*] **Wenxuan Fang**[1], **Xin Liu**[2], **Yide Qiu**[1], **Zhen Cui**[3,*]
[1]School of Computer Science and Engineering, Nanjing University of Science and Technology
[2]Nanjing Seetacloud Technology
[3]School of Artificial Intelligence, Beijing Normal University

## Abstract

Multi-domain task incremental learning (MTIL) demands models to master domain-specific expertise while preserving generalization capabilities. Inspired by human lifelong learning [1, 2], which relies on revisiting, aligning, and integrating past experiences, we propose a Learning and Ensembling Bridge Adapters (LEBA) framework. To facilitate cohesive knowledge transfer across domains, specifically, we propose a continuous-domain bridge adaptation module, leveraging the distribution transfer capabilities of Schrödinger bridge for stable progressive learning. To strengthen memory consolidation, we further propose a progressive knowledge ensemble strategy that revisits past task representations via a diffusion model and dynamically integrates historical adapters. For efficiency, LEBA maintains a compact adapter pool through similarity-based selection and employs learnable weights to align replayed samples with current task semantics. Together, these components effectively mitigate catastrophic forgetting and enhance generalization across tasks. Extensive experiments across multiple benchmarks validate the effectiveness and superiority of LEBA over state-of-the-art methods.

## 1 Introduction

Deep learning has made strides [3, 4, 5], particularly in the realm of large-scale foundation models [6, 7], with recent research further validating these advancements. However, traditional fully-supervised training methods struggle to address this challenge due to the high computational cost involved in integrating new-coming data with historical datasets. Incremental learning [8, 9], also known as continual learning, provides an effective method by incrementally learning classes, with each training task focusing solely on new-coming samples. Many methods [10, 11, 12] have actively addressed the challenges of continual learning, such as knowledge graph preservation [13], self-supervised learning [14], and replay data [15]. While these methods demonstrate potential in memorization and scalability, they mainly focus on incremental learning from batched data of a homogeneous domain.

In contrast, multi-domain task incremental learning (MTIL)—the focus of this work—aims to learn from a sequence of heterogeneous domains. The paradigm requires an effective transfer and adaptation across diverse domains while incrementally learning new ones, where catastrophic forgetting may be even more severe. Specifically, the model should not only maintain stability in retaining knowledge from previously learned domains, but also develop generalization capabilities for unseen domains, referring to the problem of **zero-shot**. Recently, vision-language models as well as knowledge distillation have been used for zero-shot MTIL. For instance, incorporating zero-shot generalization into CLIP has proven effective in mitigating knowledge degradation [16]. Further, MoE-Adapters [17] designs task-specific and task-independent components and leverages Mixture-of-Experts [18] for adaptive task learning. These methods offer promising advancements for zero-shot MTIL.

---

*Corresponding Author.

39th Conference on Neural Information Processing Systems (NeurIPS 2025).

Fundamentally, MTIL requires not only domain-specific knowledge but also the ability to capture the cross-domain transfer. A well-constructed cross-domain transfer mode ensures both the mitigation of catastrophic forgetting and better generalization in an unseen domain. In this process–akin to human learning [1, 2]–revisiting, aligning and integrating past experiences become crucial for memory consolidation. To this end, two key challenges need to be addressed: i) How to establish incremental transfer from previously learned adapters to current adapter? ii) How to replay knowledge beyond the constraints of task order and domain-specific features?

To address the above issues, in this work, we propose Learning and Ensembling Bridge Adapters (LEBA), a novel framework for multi-domain task incremental learning. LEBA introduces an incremental bridge-transfer mechanism to align the latent distributions of current and previous adapters, facilitating effective cross-domain knowledge transfer. Specifically, we design continuous-domain bridge adapters that act as incremental knowledge bridges across sequential tasks. These adapters ensure knowledge cohesion and inheritance between existing and new task domains, thereby stabilizing the incremental learning. The integrated transfer mechanism not only mitigates catastrophic forgetting effectively but also promotes progressive model optimization throughout the task sequence.

During sequential domain learning, LEBA also enhances memory consolidation by actively revisiting past experiences. Unlike traditional methods [8, 19] that rely on storing subsets of prior samples for replay– constrained by task order and data characteristics, we propose a progressive knowledge ensemble method, which could flexibly revisit prior knowledge without these constraints. By leveraging a pretrained diffusion model [20], our LEBA could reconstruct samples from any previously tasks. To optimize memory efficiency, we maintain a compact adapter pool by selectively preserving representative adapters through similarity-based matching. Furthermore, since different adapters may interpret replayed samples in distinct ways, we introduce a learnable weighting way to tailor the replay process to individual sample attributes. By adaptively integrating historical knowledge with new task adaptation, LEBA can achieve superior performance and generalization capabilities.

In summary, our primary contributions are four-fold: i) propose learning and ensembling bridge adapters framework for MTIL, facilitating knowledge transfer and mitigating catastrophic forgetting; ii) design continuous-domain bridge adaptation to align and transfer domain knowledge across sequential tasks; iii) introduce progressive knowledge ensemble regardless of task-learning sequence, enabling flexible integration of prior knowledge; iv) report state-of-the-art results on two task settings.

## 2 Related work

**Multi-domain task incremental learning:** Although the above method exhibits promising performance in incremental learning, it struggles to address a critical capability of vision-language incremental models: zero-shot transfer to unseen knowledge. In contrast to incremental learning, which centers on knowledge from a single domain, multi-domain incremental learning requires the sequential acquisition of knowledge from multiple domains. This mode necessitates that the incremental model not only incrementally learn new tasks and mitigate catastrophic forgetting but also effectively transfer knowledge across a range of diverse domains. Notably relevant is ZSCL [16], which employs parameter regularization in the incremental learning of large-scale models. Additionally, MoE-Adapters [17] enhance learning by integrating task-specific components into the CLIP model, thereby boosting its adaptability.

**Incremental learning:** Previous works in incremental learning have focused on developing a variety of architectures [21], including memory-based, regularization-based, and dynamic-based models. Memory-based methods preserve historical knowledge by storing it within a memory bank, which is periodically accessed and updated during incremental learning [19, 10, 22, 15]. Regularization-based methods integrate explicit regularization terms into the weights to mediate between previous and new-coming tasks [23, 24, 25] or data [26, 9]. Dynamic methods tackle incremental learning by progressively augmenting the baseline with new parameters, such as neurons, branches, or prediction heads [27, 28, 29, 30, 31].

**Schrödinger Bridge:** Schrödinger Bridge (SB) [32, 33] is a conditional diffusion model that solves an entropy-regularized optimal transport problem aimed at identifying the diffusion process between two distributions. Recently, Liu et al. [34] have introduced a tractable special case of dynamic stochastic bridges, which has demonstrated notable efficiency in image manipulation tasks such as image restoration and super-resolution on real-world datasets. Moreover, Schrödinger bridges belong

to a class of neural stochastic differential equations that, in contrast to diffusion models, facilitate the translation of samples across arbitrary domains with minimal transport costs. The learning of these SBs typically involves two main algorithmic approaches: flow matching [35], which distills SBs between mini-batches using optimal transport; iterative proportional fitting [36], which focus on iteratively minimizing transport costs by training models on input-output pairs generated by the models themselves. Together, these methods have enhanced the flexibility and efficiency of learning in the context of both Schrödinger bridge models [37] and broader stochastic dynamic frameworks.

## 3 The Proposed Method

### 3.1 Problem Formulation

Multi-domain task incremental learning (MTIL) involves sequentially learning from a stream of labeled task domains, where historical data becomes unavailable in subsequent stages. The goal is to evaluate not only the model's adaptability to incremental learning but also its resistance to catastrophic forgetting. Formally, given a sequence of $T$ task domains, denoted as $\{\mathcal{S}^t\}_{t=1}^T$, we want to learn an incremental model (or adapter) $\Theta^t$ based on the current task state $\mathcal{S}$ as well as previous available models. Each task domain $\mathcal{S}^t$ consists of a dataset $\mathcal{D}$ and a semantic set $\mathcal{C}$, defined as $\mathcal{S}^t := (\mathcal{D}^t, \mathcal{C}^t)$ for the $t$-th domain. The dataset $\mathcal{D}^t$ usually consists of input-label pairs, denoted as $\mathcal{D}^t := (x_i^t, y_i^t)_{i=1}^{N_t}$, where $N_t$ is the total number of samples in task $\mathcal{S}^t$. The semantic set $\mathcal{C}^t := \{c_j^t\}_{j=1}^{M_t}$ describes certain semantic information (e.g., class information $y_i^t$), with $M_t$ denoting the number of distinct class names. In this incremental paradigm, task domains are typically non-overlapping in their class labels, i.e., for any two task domains $\mathcal{S}^i, \mathcal{S}^j, \mathcal{C}^i \cap \mathcal{C}^j = \varnothing$ if $i \neq j$. A conventional solution of MTIL is to finetune the previous model via: $\Theta^t \leftarrow \Theta^{t-1} + \lambda \frac{\partial \zeta(\mathcal{S}^t)}{\partial \Theta}$, where $\zeta$ is a supervised loss function (e.g., cross entropy over class labels) and $\lambda$ is the learning rate. However, balancing new-domain adaptation with catastrophic forgetting remains a challenging problem, despite some existing efforts [16, 17] to mitigate this problem.

In contrast, we propose to learn a cross-domain adapter $\Theta$ by revisiting and aligning past knowledge. Concretely, we formulate multi-domain task incremental learning as:

$$\Theta^t \leftarrow \arg\min_{\Theta^{t-1}, \omega, \theta} \underbrace{\zeta_S(\mathcal{S}^t; \Theta^{t-1})}_{\text{supervised info.}} + \alpha \sum_{\widehat{x}_i^t \sim \mathcal{G}} \underbrace{\zeta_{\mathcal{A}}(g(\widehat{x}_i^t, c_i^t; \Theta^{t-1}), g(\widehat{x}_i^t, c_i^t; \mathcal{P}_K, \omega); \Gamma)}_{\text{knowledge alignment}}, \qquad (1)$$

where the replayed sample $\widehat{x}_i^t$ is sampled from a generator $\mathcal{G}$ conditioned on historical semantic concepts $\{\mathcal{C}^j\}_{j=1}^{t-1}$, i.e, $\widehat{x}_i^t \sim \mathcal{G}(\{\mathcal{C}^j\}_{j=1}^{t-1}; \vartheta)$; a dynamic adapter pool $\mathcal{P}_K$ of size $K$ is introduced to store useful historical adapters, i.e., $\mathcal{P}_K = \{\Theta^l\}_{l=j_1}^{j_K}$ with $j_k \in \{1, \cdots, t-1\}$; the weights $\omega$ quantify the relevance of replayed sample $\widehat{x}_i^t$ to the adapters in the pool $\mathcal{P}_K$, while $g(\cdot)$ denotes a feature extractor; the alignment loss $\zeta_{\mathcal{A}}$ over an operator $\mathcal{A}$, parameterized by $\Gamma$, measures distribution similarity between responses of current adapter and historical adapters in $\mathcal{P}_K$. By integrating supervised learning with historical knowledge alignment, our LEBA could mitigate catastrophic forgetting and enhance generalization to unseen domains–mirroring human learning processes where memory consolidation relies on revisiting past experiences.

### 3.2 Overview

Building on the formulation in Eqn. (1), our framework focuses on two key components: i) designing $\mathcal{A}$ as a cross-domain adapter and ii) dynamically integrating knowledge in $\mathcal{P}_K$. To this end, we propose Continuous-domain Bridge Adaptation (CBA) in Section 3.3 and Progressive Knowledge Ensemble (PKE) in Section 3.4. In CBA, rather than optimizing $\Theta^t$ solely via the supervised loss $\zeta_S$, we design a bridge-matching adapter $\Theta^t$ aligned with historical adapters through distribution transfer. A diffusion-based generator $\mathcal{G}$ is used to synthesize samples from the observed concept set to facilitate knowledge alignment. In PKE, we construct a dynamic buffer pool $\mathcal{P}_K$ (size $K$) to store historically significant adapters, balancing computational efficiency with knowledge retention. To address discriminability variations among adapters, we design an adaptive ensemble way with learnable weight $\omega$, ensuring both alignment and discriminative inference. Together, these components enable incremental learning to refine adapters and dynamically enhance knowledge transfer. Alongside these components, our framework incorporates a vision-language backbone with dedicated encoders for

**Algorithm 1** LEBA Training Procedure

---

**Input:** Task sequence $\mathcal{S}^t = \{(\mathcal{D}^t, \mathcal{C}^t) \mid t = 1, \ldots, T\}$; diffusion-based generator $\mathcal{G}$; continuous-domain bridge adapter $\Gamma$; adaptive weight $\omega$; incremental model $\Theta^{t=1}$
**Output:** Incremental model $\Theta^T$, adaptive weights $\omega$, and continuous-domain bridge adapter $\Gamma$
 1: Initialize model $\Theta^{t=1}$, generator $\mathcal{G}$, weight $\omega$, and adapter $\Gamma$
 2: **for** $t = 1$ to $T$ **do**
 3:     # Supervised update with current task data
 4:     Train $\Theta^{t=1}$ on $\mathcal{S}^{t=1} = (\mathcal{D}^{t=1}, \mathcal{C}^{t=1})$
 5:     **if** $t > 1$ **then**
 6:         # Progressive Knowledge Ensemble
 7:         Construct adapter pool $\mathcal{P}_K$ from previous adapters $\{\Theta^l\}_{l=j_1}^{j_K}$ with $j_k \in \{1, \cdots, t-1\}$
 8:         Generate replay samples $\widehat{x}_i^t$ from generator $\mathcal{G}$ conditioned on semantic concepts $\{\mathcal{C}^j\}_{j=1}^{t-1}$
 9:         Compute adaptive weights $\omega$ for each replay sample $\widehat{x}_i^t$
10:         Evaluate similarity $\eta$ of current adapter $\Theta^t$ and update adapter pool $\mathcal{P}_K$
11:         # Continuous-Domain Bridge Adaptation
12:         Construct continuous-domain bridge adapter $\Gamma$ with replay data $\widehat{x}_i^t$ and adaptive weight $\omega$
13:         # Joint Optimization
14:         Update $\Theta^t$, $\omega$, and $\Gamma$ by minimizing the total loss $\zeta_{\text{total}}$ (Eqn. 12)
15:     **end if**
16:     # Update the incremental model for the next domain
17:     $\Theta^{t+1} \leftarrow \Theta^t$
18: **end for**

---

processing image and semantic information. Specifically, we extract both image and text features for label prediction by reformulating the feature extractor $g$ as a decomposing form:

$$g(x_i^t, c_i^t, \Theta^t) := g_{img}(x_i^t, \Theta_{img}^t) \otimes g_{txt}(c_i^t, \Theta_{txt}^t), \tag{2}$$

where $g_{img}$ and $g_{txt}$ denote the image and text encoders, respectively. $\otimes$ represents the element-wise product. Following common practice, we initialize these encoders using pre-trained models (e.g., CLIP [38]) as backbones, with additional adaptation layers learned for task-specific fine-tuning. Please note the adapter parameters that are structured as $\Theta^t = \{\Theta_{img}^t, \Theta_{txt}^t\}$. The subsequent subsections elaborate on the details of CBA and PKE, followed by the LEBA training optimization. The LEBA training process is shown in Algorithm 1.

### 3.3 Continuous-Domain Bridge Adaptation

The adapter requires not only domain-specific adaptation capabilities but also the ability to facilitate cross-domain knowledge transfer. Effective knowledge transfer should simultaneously mitigate catastrophic forgetting while improving generalization performance on unseen domains. To achieve this, we leverage the Schrödinger Bridge (SB) mechanism to facilitate inter-task knowledge transfer by aligning probability distributions between current and previous adapters. Specifically, we design a continuous-domain bridge adapter for cross-domain distribution transfer.

Given a sample $\widehat{x}_i$, the feature $Z_1 = g(\widehat{x}_i, c_i, \Theta^t) \sim P_1$ encoded by the current adapter $\Theta^t$ follows the probability distribution $P_1$. Similarly, for a historical adapter in the buffer pool $\mathcal{P}_K$, $\Theta^{j_k} \in \mathcal{P}_K$ (where $j_k \leq t - 1$), the feature $Z_0 = g(\widehat{x}_i, c_i, \Theta^{j_k}) \sim P_0$ can be obtained. The continuous-domain bridge adapter can be formalized as:

$$\begin{aligned}
dZ_m &= [f_m + \beta_m \nabla \log \Psi(Z_m, m)]dm + \sqrt{\beta_m}dW_m, \quad Z_0 \sim P_0, \\
dZ_m &= [f_m - \beta_m \nabla \log \hat{\Psi}(Z_m, m)]dm + \sqrt{\beta_m}d\overline{W}_m, \quad Z_1 \sim P_1,
\end{aligned} \tag{3}$$

where $P_0$ and $P_1$ denotes the source and target distributions, $m$ denotes the time-step, $\{W_m, \overline{W}_m\}$ refer to the standard Wiener process and its time reversal and $\{f_m, \beta_m\}$ are the drift and diffusion coefficients. The pair of functions $\{\Psi, \hat{\Psi}\}$ is said to solve the following coupled PDEs. The Eqn.(3) and its time-reversal are directly derived from the Fokker-Planck equation [39] corresponding to the SDE in Eqn.(4), as follows:

$$\begin{aligned}
dZ_m &= f_m dm + \sqrt{\beta_m}dW_m, \quad Z_0 \sim \hat{\Psi}(\cdot, 0), \\
dZ_m &= f_m dm + \sqrt{\beta_m}d\overline{W}_m, \quad Z_1 \sim \Psi(\cdot, 1).
\end{aligned} \tag{4}$$

Since $\Psi$ and $\hat{\Psi}$ contain complex drift terms that are difficult to compute, we simplify a certain form for the boundary distributions $P_0$ and $P_1$. We define the energy potential functions as $\hat{\Psi}(\cdot, 0) = P_0(\cdot) := \delta_a(\cdot)$ and $\Psi(\cdot, 1) = P_1(\cdot)/\hat{\Psi}(\cdot, 1)$, where $\delta_a(\cdot)$ is the Dirac delta distribution centered on $a \in \mathbb{R}$. This choice ensures that the diffusion process becomes computationally manageable.

Consequently, we can approximate both the forward and backward with the following Gaussian posterior as:

$$Z_m \sim q(Z_m|Z_0, Z_1) = \mathcal{N}(Z_m; \mu_m, \Sigma_m),$$

$$\text{s.t. } \mu_m = \frac{\overline{\sigma}_m^2}{\overline{\sigma}_m^2 + \sigma_m^2} Z_0 + \frac{\sigma_m^2}{\overline{\sigma}_m^2 + \sigma_m^2} Z_1, \ \Sigma_m = \frac{\overline{\sigma}_m^2 \sigma_m^2}{\overline{\sigma}_m^2 + \sigma_m^2}, \tag{5}$$

where $\sigma_m^2 = \int_0^m \beta_{m'} \mathrm{d}m'$ and $\overline{\sigma}_m^2 = \int_m^1 \beta_{m'} \mathrm{d}m'$ represent the cumulative noise variances in the forward and backward directions. We take $P(Z_0, Z_1) = P_0(Z_0)P_1(Z_1|Z_0)$ and $f = 0$, and construct tractable SB between individual knowledge distribution from $Z_0$ and $P_1(Z_1|Z_0)$.

Based on the adapter formulation, we derive an approximate reverse SDE from Eqn.(3) to simulate the transfer from $Z_1$ to $Z_0$ by estimating the score function $\log \hat{\Psi}(Z_m, m|\hat{x}_i, c_i) = \varepsilon(\hat{x}_i, c_i, Z_m, m; \Gamma)/\beta_m$, formally:

$$\mathrm{d}Z_m = (\beta_m/\sigma_m)\varepsilon(\hat{x}_i, c_i, Z_m, m; \Gamma)\mathrm{d}m + \sqrt{\beta_m}\mathrm{d}W_m, \tag{6}$$

where $P_1$ denotes the distribution of $Z_1$, i.e., $Z_1 \sim P_1(Z_1|\hat{x}_i, c_i)$, and $\varepsilon$ is a continuous-domain bridge realized through a neural network parameterized by $\Gamma$. The adapter network is optimized to approximate the score function $\nabla_Z \log p_m(Z_m|\hat{x}_i, c_i)$ by minimizing the following objective function:

$$\zeta_{CBA} = \mathbb{E}_{\hat{x}_i, c_i, Z_m} \left[ \left\| \varepsilon(\hat{x}_i, c_i, Z_m, m; \Gamma) - \frac{Z_m - Z_0}{\sigma_m} \right\|_2^2 \right], \tag{7}$$

where $m \in \mathcal{U}([0, 1])$ and $Z_m \sim q(Z_m|Z_0, Z_1)$ are defined in Eqn.(5).

For the above formula, we propose to construct a continuous-domain bridge adapter to connect the previous distribution (from prior adapters) to the current distribution (from the new adapter). This enables the incremental model to mitigate catastrophic forgetting while optimizing performance effectively. By facilitating smooth transitions between past and present tasks, our approach offers a novel framework for understanding and implementing incremental learning.

### 3.4 Progressive Knowledge Ensemble

The absence of historical samples presents a fundamental challenge to revisiting prior knowledge in MTIL. Existing methods [19, 8] typically preserve features from selected past samples for replay, but remain constrained by task order and data-specific dependencies, thereby limiting their robustness against catastrophic forgetting. To overcome this limitation, we draw inspiration from human learning [1, 2] and propose a progressive knowledge ensemble that enables flexible knowledge reuse.

To enable flexible knowledge reuse, we dynamically maintain an adapter pool $\mathcal{P}_K$ of size $K$ to store useful historical adapters. For each new adapter $\Theta^t$, we measure its similarity to those in the pool and update the pool by replacing the least representative one when necessary. Formally, the similarity between the current adapter and the $j$-th adapter in the pool is computed as:

$$\eta_j = D_{KL}(g(x_i^t, c_i^t, \Theta^t)||g(x_i^t, c_i^t, \Theta^j)), \text{s.t. } j = 1, 2, ..., K, \tag{8}$$

where $\eta_j$ denotes the similarity between the current adapter and the $j$-th adapter in the dynamic adapter pool, computed over the current domain samples using KL divergence [40]. A threshold-based strategy determines whether the current adapter should replace an existing one.

How can replay samples be obtained without relying on traditional methods such as storing sample features or task-sequential replay? Inspired by human learning—where individuals can recognize whether a concept has been previously encountered—we propose an alternative strategy. Specifically, each replayed sample $\hat{x}_i^t$ is drawn from a diffusion-based generator $\mathcal{G}$ conditioned on previously encountered semantic concepts $\{\mathcal{C}^j\}_{j=1}^{t-1}$, i.e, $\hat{x}_i^t \sim \mathcal{G}(\{\mathcal{C}^j\}_{j=1}^{t-1}; \vartheta)$, where $\vartheta$ denotes the generator parameters. Let the replay set be denoted as $\hat{X}^t$, consisting of $|\hat{X}^t|$ samples, i.e., $\hat{X}^t = \{\hat{x}_i^t\}_{i=1}^{|\hat{X}^t|}$.

Meanwhile, an learnable weight $\omega$ is computed for each replay data $\widehat{x}_i^t$, which can be formulated as:

$$\zeta_{PKE} = 1 - \cos(\sum_{j=1}^{k} \omega_j \cdot g_{img}(\widehat{x}_i^t, \Theta_{img}^j); g_{txt}(c_i^t, \Theta_{txt}^j)), \text{s.t. } \omega_j = \omega_1, \omega_2, ..., \omega_k, \qquad (9)$$

where $\omega_j$ represents the similarity between the replay sample $\widehat{x}_i^t$. Through this process, we obtain an adaptive weight $\omega$ to evaluate the similarity between the replay sample $\widehat{x}_i$ and the previous adapters, thereby providing a more accurate previous distribution to construct the Schrödinger bridge, further enhancing knowledge transfer between current and previous adapters.

### 3.5 Optimizing the LEBA

To define the loss function $\zeta_{total}$ in the above process, we can utilize replay data $\widehat{x}_i$ (see Section 3.4) and the constructed continuous-domain bridge adapter (see Section 3.3) to facilitate knowledge transfer and integration between adapters. Formally, considering that each adapter has a different understanding of the replay data $\widehat{x}_i$ in the $k$-th(i.e., $k > 1$) incremental task, its probability distribution can be rewritten as:

$$\widehat{Z}_0 = \sum_{j=0}^{k} \omega_j \cdot (g_{img}(\widehat{x}_i, \Theta_{img}^j) \otimes g_{txt}(c_i, \Theta_{txt}^j)), \qquad (10)$$

where $\widehat{Z}_0$ denotes the integrated predictive distribution over all adapters for the replay sample $\widehat{x}_i$. Consequently, the final objective loss for the continuous-domain bridge adapter, based on Eqn.(7), can be reformulated as follows:

$$\zeta_{RCBA} = \mathbb{E}_{\widehat{x}_i, c_i, Z_m} \left[ \left\| \varepsilon(\widehat{x}_i, c_i, Z_m, m; \Gamma) - \frac{Z_m - \widehat{Z}_0}{\sigma_m} \right\|_2^2 \right]. \qquad (11)$$

All components are ultimately integrated into the unified LEBA framework, thereby preserving the learned knowledge distribution across tasks. The final optimization objective can be formally defined as:

$$\zeta_{total} = \zeta_{CE}(\Theta^t) + \gamma\zeta_{RCBA}(\Gamma) + \beta\zeta_{PKE}(\omega), \qquad (12)$$

where $\gamma$ and $\beta$ are balance factors. We construct a continuous-domain bridge adapter between different adapters by replaying samples (instead of stage-wise replay). Our LEBA effectively mitigates catastrophic forgetting by maintaining the probability distribution of the learned knowledge. Furthermore, our LEBA not only facilitates knowledge transfer across domains but also enables the incremental model to retain knowledge of previous tasks while learning new ones.

## 4 Experiment

### 4.1 Experimental Setting

**Datasets:** We evaluate our LEBA in the multi-domain task incremental learning(MTIL) [16]. In this configuration, tasks are sourced from multiple domains, each necessitating unique domain knowledge to achieve high accuracy. The MTIL benchmark comprises 11 tasks and contains a total of 1,201 classes. We evaluate the method using two different task orders: the first follows an alphabetical order (Order-I): Aircraft [41], Caltech101 [42], CIFAR100 [43], DTD [44], EuroSAT [45], Flowers [46], Food [47], MNIST [48], OxfordPet [49], StanfordCars [50], and SUN397 [51]. The second uses a random order (Order-II): StanfordCars, Food, MNIST, OxfordPet, Flowers, SUN397, Aircraft, Caltech101, DTD, EuroSAT, and CIFAR100. By default, experiments are conducted using Order-I.

**Evaluation Metrics:** To evaluate LEBA in the multi-task incremental learning (MTIL) setting, we follow the protocol introduced in ZSCL [16], which includes three metrics: "Transfer", "Last", and "Average". "Transfer" measures the model's zero-shot generalization to unseen tasks, while "Last" evaluates its ability to retain knowledge from previous tasks. "Average" captures overall performance by averaging the results of "Transfer" and "Last". However, these metrics do not explicitly quantify the extent of forgetting across tasks. To address this, we introduce a new "Preserve" metric, which captures forgetting dynamics by analyzing the lower triangular portion of the accuracy matrix. Formally, "Preserve" is defined as Preserve $= \frac{1}{T(T-1)/2} \sum_{i=1}^{T} \sum_{j=1}^{i-1} \text{acc}_{i,j}$, where $\text{acc}_{i,j}$ denotes

Table 1: Comparison with state-of-the-art methods on the multi-domain task incremental learning benchmark (Order-I) in terms of "Transfer", "Average", "Last" and "Preserve" scores (%).

| | Method | Aircraft | Caltech101 | CIFAR100 | DTD | EuroSAT | Flowers | Food | MNIST | OxfordPet | Cars | SUN397 | *Average* |
|---|---|---|---|---|---|---|---|---|---|---|---|---|---|
| CLIP | Zero-shot | 24.3 | 88.4 | 68.2 | 44.6 | 54.9 | 71.0 | 88.5 | 59.4 | 89.0 | 64.7 | 65.2 | 65.3 |
| | Full Fine-tune | 62.0 | 95.1 | 89.6 | 79.5 | 98.9 | 97.5 | 92.7 | 99.6 | 94.7 | 89.6 | 81.8 | 89.2 |
| Transfer | Continual-FT | | 67.1 | 46.0 | 32.1 | 35.6 | 35.0 | 57.7 | 44.1 | 60.8 | 20.5 | 46.6 | 44.6 |
| | LwF [9] | | 74.5 | 56.9 | 39.1 | 51.1 | 52.6 | 72.8 | 60.6 | 75.1 | 30.3 | 55.9 | 58.9 |
| | iCaRL [57] | | 56.6 | 44.6 | 32.7 | 39.3 | 46.6 | 68.0 | 46.0 | 77.4 | 31.9 | 60.5 | 50.4 |
| | WiSE-FT [58] | | 73.5 | 55.6 | 35.6 | 41.5 | 47.0 | 68.3 | 53.9 | 69.3 | 26.8 | 51.9 | 52.3 |
| | ZSCL [16] | | 86.0 | 67.4 | **45.4** | 50.4 | 69.1 | 87.6 | **61.8** | 86.8 | 60.1 | **66.8** | 68.1 |
| | MoE-Adapters [17] | | 87.9 | 68.2 | 44.4 | 49.9 | **70.7** | **88.7** | 59.7 | **89.1** | 64.5 | 65.5 | 68.9 |
| | Ours | | **88.5** | **68.3** | 44.8 | 49.4 | 70.2 | 88.6 | 60.9 | **89.1** | **64.8** | 64.2 | **69.2(+0.3)** |
| Average | Continual-FT | 25.5 | 81.5 | 59.1 | 53.2 | 64.7 | 51.8 | 63.2 | 64.3 | 69.7 | 31.8 | 49.7 | 55.9 |
| | LwF [9] | 36.3 | 86.9 | 72.0 | 59.0 | 73.7 | 60.0 | 73.6 | 74.8 | 80.0 | 37.3 | 58.1 | 64.7 |
| | iCaRL [57] | 35.5 | 89.2 | 72.2 | 60.6 | 68.8 | 70.0 | 78.2 | 62.3 | 81.8 | 41.2 | 62.5 | 65.7 |
| | WiSE-FT [58] | 26.7 | 86.5 | 64.3 | 57.1 | 65.7 | 58.7 | 71.1 | 70.5 | 75.8 | 36.9 | 54.6 | 60.7 |
| | ZSCL [16] | 45.1 | 92.0 | 80.1 | 64.3 | 79.5 | 81.6 | **89.6** | **75.2** | 88.9 | 64.7 | **68.0** | 75.4 |
| | MoE-Adapters [17] | 50.2 | 91.9 | 83.1 | 69.4 | 78.9 | 84.0 | 89.1 | 73.7 | 89.3 | 67.7 | 66.9 | 76.7 |
| | Ours | **53.9** | **94.9** | **83.8** | **70.8** | **79.8** | **85.1** | 89.1 | 74.8 | **89.3** | **69.2** | 65.8 | **77.9(+1.2)** |
| Last | Continual-FT | 31.0 | 89.3 | 65.8 | 67.3 | 88.9 | 71.1 | 85.6 | 99.6 | 92.9 | 77.3 | 81.1 | 77.3 |
| | LwF [9] | 26.3 | 87.5 | 71.9 | 66.6 | 79.9 | 66.9 | 83.8 | 99.6 | 92.1 | 66.1 | 80.4 | 74.6 |
| | iCaRL [57] | 35.8 | 93.0 | 77.0 | 70.2 | 83.3 | 88.5 | 90.4 | 86.7 | 93.2 | 81.2 | 81.9 | 80.1 |
| | WiSE-FT [58] | 27.2 | 90.8 | 68.0 | 68.9 | 86.9 | 74.0 | 87.6 | **99.6** | 92.6 | 77.8 | 81.3 | 77.7 |
| | ZSCL [16] | 40.6 | 92.2 | 81.3 | 70.5 | 94.8 | 90.5 | **91.9** | 98.7 | **93.9** | 85.3 | 80.2 | 83.6 |
| | MoE-Adapters [17] | 49.8 | 92.2 | 86.1 | 78.1 | 95.7 | 94.3 | 89.5 | 98.1 | 89.9 | 81.6 | 80.0 | 85.0 |
| | Ours | **55.1** | **95.2** | **87.4** | **78.8** | **97.2** | **97.3** | 89.5 | 99.1 | 89.6 | **88.8** | **82.4** | **87.3(+2.3)** |
| Preserve | Continual-FT | 29.2 | 87.2 | 61.5 | 63.2 | 84.4 | 68.5 | 80.6 | 96.2 | 88.3 | 74.2 | | 71.3 |
| | LwF [9] | 25.4 | 84.4 | 69.3 | 62.4 | 75.2 | 63.8 | 79.5 | 97.4 | 89.5 | 63.2 | | 69.8 |
| | iCaRL [57] | 30.5 | 91.1 | 74.6 | 66.4 | 79.2 | 83.1 | 86.5 | 82.1 | 89.4 | 76.2 | | 74.8 |
| | WiSE-FT [58] | 26.2 | 85.6 | 62.1 | 63.2 | 82.3 | 75.1 | 77.2 | 97.5 | 90.4 | 79.2 | | 74.9 |
| | ZSCL [16] | 44.1 | 92.5 | 82.5 | 70.7 | 95.9 | 91.2 | **91.9** | 98.8 | **94.2** | 85.3 | | 80.0 |
| | MoE-Adapters [17] | 50.0 | 92.3 | 86.3 | 78.8 | 95.4 | 95.0 | 89.5 | 98.2 | 89.8 | 81.6 | | 82.4 |
| | Ours | **53.7** | **95.4** | **87.0** | **80.4** | **96.9** | **97.4** | 89.5 | **99.1** | 89.6 | **88.8** | | **84.5(+2.1)** |

the accuracy on $j$-th domain after training on $i$-th task. This metric provides a more comprehensive assessment of the model's ability to preserve learned knowledge in the MTIL.

**Implementation Details:** Following previous work [16], we adopt CLIP with ViT-B/16 [52] as the backbone for all experiments. Each task's adapter is composed using LoRA [53]. For generative replay, we employ the Stable Diffusion-V1.4 model [54], capable of generating samples that closely approximate the original data in both fidelity and discriminative quality. The continuous-domain bridge adapter $\Gamma$ is implemented as a four-layer MLP. We set the balancing factors $\gamma = 0.1$ and $\beta = 0.4$, and use a step size of $m = 20$ and an adapter selection threshold of $\eta = 0.3$, and an adapter pool containing $K = 2$ adapters. Optimization is performed using the AdamW optimizer [55], with label smoothing [56] applied to improve baseline performance. For the MTIL benchmark, we use a batch size of 64 and search the learning rate $\alpha$ within $\{1 \times 10^{-3}, \ldots, 1 \times 10^{-5}\}$. All experiments are conducted using PyTorch on NVIDIA GeForce RTX 4090 GPUs.

## 4.2 Comparison with State-of-the-art Methods

Table 1 presents the detailed results of the "Transfer", "Avg", "Last", and "Preserve" metrics on the MTIL benchmark across all evaluated methods and datasets. Zero-shot refers to the prediction performance of the initial CLIP model without any task-specific adaptation, while Fine-tune represents the accuracy achieved by fully fine-tuning on each dataset, serving as an upper bound in the absence of forgetting. The results reveal that both zero-shot prediction and newly learned knowledge suffer from performance degradation under incremental learning. While existing methods partially mitigate this issue, they generally fail to preserve strong zero-shot capabilities. Our proposed method, LEBA (denoted as Ours), consistently outperforms the strongest MoE-Adapter across most tasks, demonstrating superior overall performance and a more favorable stability–plasticity trade-off. Further validation shows that the CBA module, by constructing a continuous-domain bridge adapter, effectively integrates and revisits previously learned knowledge while adapting to new domains; by coupling supervised learning with historical knowledge alignment (e.g., feature/adapter consistency), the model enables smooth knowledge transition and durable retention. In parallel, the PKE module addresses task-order limitations, enabling flexible reuse of prior knowledge regardless of the incremental sequence via progressive knowledge ensemble and selective routing. These components mitigate catastrophic forgetting and enhance zero-shot generalization to unseen categories.

Table 2: Comparison with state-of-the-art methods on the multi-domain task incremental learning benchmark (Order-II) in terms of "Transfer", "Average", "Last" and "Preserve" scores (%).

| | Method | Cars | Food | MNIST | OxfordPet | Flowers | SUN397 | Aircraft | Caltech101 | DTD | EuroSAT | CIFAR100 | *Average* |
|---|---|---|---|---|---|---|---|---|---|---|---|---|---|
| CLIP | Zero-shot | 64.7 | 88.5 | 59.4 | 89.0 | 71.0 | 65.2 | 24.3 | 88.4 | 44.6 | 54.9 | 68.2 | 65.3 |
| | Full Fine-tune | 89.6 | 92.7 | 99.6 | 94.7 | 97.5 | 81.8 | 62.0 | 95.1 | 79.5 | 98.9 | 89.6 | 89.2 |
| Transfer | Continual-FT | | 85.9 | 59.6 | 57.9 | 40.0 | 46.7 | 11.1 | 70.0 | 30.5 | 26.6 | 37.7 | 46.6 |
| | LwF [9] | | 87.8 | 58.5 | 71.9 | 46.6 | 57.3 | 12.8 | 81.4 | 34.5 | 34.5 | 46.8 | 53.2 |
| | iCaRL [57] | | 86.1 | 51.8 | 67.6 | 50.4 | 57.9 | 11.0 | 72.3 | 31.2 | 32.7 | 48.1 | 50.9 |
| | WiSE-FT [58] | | 87.2 | 57.6 | 67.0 | 45.0 | 54.0 | 12.9 | 78.6 | 35.5 | 28.4 | 44.3 | 51.1 |
| | ZSCL [16] | | 88.3 | 57.5 | 84.7 | 68.1 | 64.8 | **21.1** | **88.2** | **45.3** | 55.2 | **68.2** | 64.1 |
| | MoE-Adapters [17] | | **88.8** | 59.5 | 89.1 | 69.9 | 64.4 | 18.1 | 86.9 | 43.7 | 54.6 | **68.2** | 64.3 |
| | Ours | | 88.7 | 60.2 | 89.3 | 71.1 | 65.1 | 18.4 | 88.5 | 45.9 | 55.3 | 68.1 | 65.1(+0.8) |
| Average | Continual-FT | 42.1 | 70.5 | **92.2** | 80.1 | 54.5 | 59.1 | 19.8 | 78.3 | 41.0 | 38.1 | 42.3 | 56.2 |
| | LwF [9] | 49.0 | 77.0 | 92.1 | 85.9 | 66.5 | 67.2 | 20.9 | 84.7 | 44.6 | 45.5 | 50.5 | 62.2 |
| | iCaRL [57] | 52.0 | 75.9 | 77.4 | 74.6 | 58.4 | 59.3 | 11.7 | 79.6 | 42.1 | 43.2 | 51.7 | 56.9 |
| | WiSE-FT [58] | 52.6 | 79.3 | 91.9 | 83.9 | 63.4 | 65.2 | 23.3 | 83.7 | 45.4 | 40.0 | 48.2 | 61.5 |
| | ZSCL [16] | 81.7 | **91.3** | 91.1 | 91.0 | 82.9 | 72.5 | 33.6 | 89.7 | 53.3 | **62.8** | 69.9 | 74.5 |
| | MoE-Adapters [17] | 84.9 | 89.9 | 89.3 | 89.4 | 86.2 | 72.2 | 33.4 | 89.4 | 53.3 | 61.4 | 69.9 | 74.7 |
| | Ours | **86.0** | 88.9 | 92.1 | **91.9** | **87.2** | **72.8** | **33.9** | **90.9** | 54.7 | 62.7 | **70.1** | 75.6(+0.9) |
| Last | Continual-FT | 24.0 | 67.3 | 99.1 | 87.4 | 44.3 | 67.0 | 29.5 | 92.3 | 61.3 | 81.0 | **88.1** | 67.4 |
| | LwF [9] | 34.6 | 69.6 | 99.3 | 88.7 | 61.1 | 72.5 | 32.5 | 88.1 | 65.6 | 90.9 | 87.9 | 71.9 |
| | iCaRL [57] | 46.0 | 81.5 | 91.3 | 82.8 | 66.5 | 72.2 | 16.3 | 91.6 | 68.1 | 83.2 | 87.8 | 71.6 |
| | WiSE-FT [58] | 35.6 | 76.9 | **99.5** | 89.1 | 62.1 | 71.8 | 27.8 | 90.8 | 67.0 | 85.6 | 87.6 | 72.2 |
| | ZSCL [16] | 78.2 | **91.1** | 97.6 | 92.5 | 87.4 | 78.2 | 45.0 | 92.3 | 72.7 | **96.2** | 86.3 | 83.4 |
| | MoE-Adapters [17] | **84.1** | 88.5 | 94.0 | 91.8 | 94.1 | 77.8 | **50.4** | 93.3 | 77.1 | 87.7 | 86.6 | 84.1 |
| | Ours | 86.2 | 88.9 | 99.2 | **93.0** | 96.5 | 79.2 | 50.1 | 95.2 | 78.2 | 95.9 | 88.1 | 86.4(+2.3) |
| Preserve | Continual-FT | 40.5 | 68.1 | 89.1 | 77.8 | 51.4 | 56.7 | 18.4 | 76.2 | 39.8 | 35.4 | | 55.3 |
| | LwF [9] | 47.4 | 76.1 | 90.1 | 83.6 | 63.8 | 64.5 | 17.2 | 80.7 | 41.8 | 43.4 | | 60.7 |
| | iCaRL [57] | 49.2 | 74.3 | 75.6 | 71.2 | 55.7 | 57.6 | 10.3 | 76.8 | 39.5 | 40.8 | | 55.1 |
| | WiSE-FT [58] | 49.7 | 76.8 | 90.4 | 81.6 | 61.2 | 63.2 | 20.4 | 80.6 | 41.2 | 38.1 | | 60.3 |
| | ZSCL [16] | 77.1 | **89.2** | 95.1 | 90.2 | 85.6 | 77.5 | 42.9 | 90.6 | 72.1 | 94.2 | | 81.5 |
| | MoE-Adapters [17] | 81.2 | 87.6 | 97.5 | 85.1 | 90.9 | 74.1 | 48.2 | 91.4 | 74.1 | **97.1** | | 82.9 |
| | Ours | **85.4** | 88.2 | **98.8** | 92.3 | 96.2 | 78.6 | 49.6 | 94.2 | 78.7 | 95.6 | | 85.8(+2.9) |

Table 3: Performance comparison of CBA module and PKE module of LEBA

| | Method | Aircraft | Caltech101 | CIFAR100 | DTD | EuroSAT | Flowers | Food | MNIST | OxfordPet | Cars | SUN397 | *Average* |
|---|---|---|---|---|---|---|---|---|---|---|---|---|---|
| **Transfer** | Baseline | | 87.2 | 67.1 | 43.3 | 48.4 | 68.5 | 86.9 | 57.1 | 87.7 | 63.2 | 63.2 | 67.3 |
| | +CBA | | 88.1 | 67.6 | 44.1 | 49.1 | 69.4 | 87.5 | 58.2 | 88.6 | 64.1 | 64.3 | 68.1 |
| | +CBA+PKE | | 88.5 | 68.3 | 44.8 | 49.4 | 70.2 | 88.6 | 60.9 | 89.1 | 64.8 | 64.4 | 69.2 |
| **Average** | Baseline | 52.8 | 93.3 | 81.2 | 68.7 | 77.9 | 82.1 | 88.1 | 73.9 | 89.1 | 66.9 | 63.8 | 76.1 |
| | +CBA | 53.5 | 94.3 | 83.2 | 69.6 | 78.4 | 84.5 | 88.4 | 74.1 | 88.6 | 68.7 | 64.4 | 77.2 |
| | +CBA+PKE | 53.9 | 94.9 | 83.8 | 70.8 | 79.8 | 85.1 | 89.1 | 74.8 | 89.3 | 69.2 | 65.8 | 77.9 |
| **Last** | Baseline | 52.1 | 93.8 | 84.3 | 76.6 | 95.3 | 95.2 | 86.7 | 97.1 | 89.1 | 84.2 | 78.1 | 84.7 |
| | +CBA | 54.2 | 94.4 | 86.8 | 78.3 | 96.3 | 96.4 | 87.5 | 98.3 | 89.2 | 87.1 | 79.6 | 86.2 |
| | +CBA+PKE | 55.1 | 95.2 | 87.4 | 78.8 | 97.2 | 97.3 | 89.5 | 99.1 | 89.6 | 88.8 | 82.4 | 87.3 |
| **Preserve** | Baseline | 52.3 | 93.5 | 86.1 | 77.4 | 94.9 | 94.8 | 87.7 | 98.1 | 88.4 | 85.2 | | 82.5 |
| | +CBA | 53.4 | 94.2 | 86.7 | 78.2 | 95.8 | 95.6 | 88.4 | 98.6 | 88.7 | 86.1 | | 83.6 |
| | +CBA+PKE | 53.7 | 95.4 | 87.0 | 80.4 | 96.9 | 97.4 | 89.5 | 99.1 | 89.4 | 88.7 | | 84.5 |

## 4.3 Ablation Study

This section focuses on analyzing the effectiveness of the proposed LEBA method. All experiments are conducted in a multi-domain task incremental learning setting, with additional analysis available in the supplementary material.

**Effectiveness of different modules:** We conduct experiments to assess the effectiveness of the proposed CBA and PKE, with detailed results shown in Table 3. The results clearly demonstrate that incorporating the CBA module consistently improves performance over the baseline, highlighting its effectiveness in enhancing cross-task knowledge adaptation and generalization. Furthermore, the integration of the PKE module leads to additional gains across all evaluation metrics, particularly in preserving prior knowledge and maintaining strong performance on the most recent tasks. This indicates that the PKE module plays a crucial role in mitigating catastrophic forgetting while enabling forward transfer. It should be noted that the PKE module cannot be used alone. Overall, the synergistic effect of CBA and PKE contributes to stable and consistent improvements, validating the robustness of the proposed LEBA architecture in MTIL.

**T-SNE visualization analysis:** We present t-SNE visualizations of ZSCL, MoE-Adapter, and LEBA (Ours) on the Flowers and Aircraft tasks, using the final models obtained after completing all incremental sessions, as shown in Fig. 1. From a visual perspective, LEBA exhibits more distinct

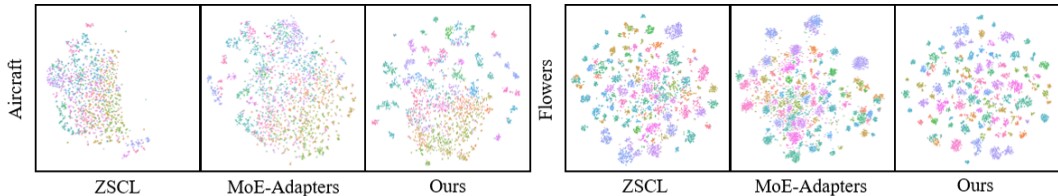

Figure 1: The t-SNE visualizations illustrate the representation evolution across multi-domain task-incremental learning sessions for various methods on two benchmark datasets.

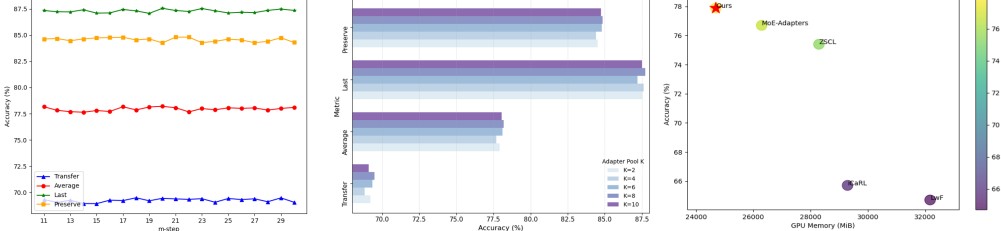

Figure 2: The ablation experiments on the $m$-step.

Figure 3: The ablation experiments on the adapter pool $K$.

Figure 4: The ablation experiments of computational cost.

class-wise separability compared to other baselines, suggesting improved representation stability and task-specific disentanglement. Notably, LEBA demonstrates stronger robustness in multi-domain continual learning, effectively consolidating previously acquired knowledge while flexibly adapting to novel tasks. These results highlight LEBA's advantage in achieving both knowledge retention and forward transfer in complex task-incremental scenarios.

**The quantity of $m$-step:** To assess the impact of the iterative step size $m$ on adapter integration in the LEBA framework, we plot the accuracy trends of "Transfer", "Average", "Last", and "Preserve" metrics as a function of $m$-step as shown in Fig. 2. Across all metrics, the performance remains remarkably stable with increasing $m$, exhibiting minimal fluctuation. This consistency suggests that the model effectively preserves previously acquired knowledge while maintaining robust performance throughout the incremental learning process. The observed stability underscores the adaptability and robustness of our LEBA framework in mitigating catastrophic forgetting and sustaining high learning capacity, particularly in retaining and transferring knowledge across long sequences of tasks in MTIL.

**Effectiveness of adapters pool size $K$:** To investigate how the size of the adapter pool influences performance and resource efficiency in LEBA, we evaluate the model under varying values of $K$. As shown in Fig. 3, increasing the adapter pool size $K$ yields only marginal changes across all four evaluation metrics, with the model maintaining consistently high performance under different settings. This phenomenon is consistent with observations in Mixture-of-Experts models, where an excessive number of experts may lead to performance degradation [59], indicating that LEBA is robust to the choice of pool size and does not depend on retaining a large number of adapters to sustain its effectiveness. Given the negligible performance improvement beyond $K = 2$, and considering the trade-off between memory overhead and model complexity, we select $K = 2$ as the default configuration to ensure efficient memory usage.

**Computational cost:** To evaluate the computational efficiency of LEBA, we compare the "Average" and memory consumption across different methods, as shown in Fig. 4. Compared to existing approaches, LEBA achieves higher accuracy with significantly lower memory usage. This indicates that our method not only improves performance but also offers superior efficiency in resource utilization. These results further highlight the effectiveness of LEBA in multi-task incremental learning, demonstrating its advantage in balancing accuracy and computational cost.

## 5 Conclusion

In this paper, we propose a novel LEBA framework designed to mitigate catastrophic forgetting in multi-domain task-incremental learning. A core component of LEBA is the continuous-domain bridge adaptation to establish a stable transfer pathway between adapters, effectively aligning the distributions of previous and current tasks. Furthermore, our progressive knowledge ensemble departs from traditional task-replay paradigms by removing the dependency on task-learning order, allowing the model to revisit and integrate prior knowledge flexibly. Extensive experiments validate the effectiveness of LEBA in enhancing both knowledge retention and transfer. In future work, we plan to extend LEBA to broader AI domains beyond vision-language tasks.

# 6 Acknowledgement

This work was supported by the National Natural Science Foundation of China (Grant Nos. 62372238 and 62476133) and the National Science and Technology Major Project of China (Grant No. 2024ZD0524600) and the Fundamental Research Funds for the Central Universities (Grant No. 11300-312200502507).

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

# A Proof of Continuous-Domain Bridge Adaptation

To support our continuous-domain bridge adaptation, we draw on the theory of Schrödinger bridges and their connection to stochastic differential equations (SDEs). Specifically, we model knowledge transition as a bidirectional diffusion process governed by forward and reverse SDEs, whose densities evolve according to the Fokker-Planck equation. By conditioning on the samples' representations, we obtain a time-dependent posterior that aligns with the solution of a Schrödinger bridge.

We begin by recalling that the density evolution of an Itô process is governed by the following stochastic differential equations (SDEs):

$$\mathrm{d}Z_m = [f_m + \beta_m \nabla \log \Psi(Z_m, m)]\mathrm{d}m + \sqrt{\beta_m}\mathrm{d}W_m, \quad Z_0 \sim p_0,$$
$$\mathrm{d}Z_m = [f_m - \beta_m \nabla \log \hat{\Psi}(Z_m, m)]\mathrm{d}m + \sqrt{\beta_m}\mathrm{d}\overline{W}_m, \quad Z_1 \sim p_1, \tag{13}$$

where these SDEs correspond to a forward and reverse Schrödinger bridge process and are described by the Fokker-Planck equation [39]:

$$\frac{\partial p(z, m)}{\partial m} = -\nabla \cdot (f_m\, p) + \frac{1}{2}\beta_m \Delta p, p(z, 0) = p_0(z). \tag{14}$$

We suggest that the PDE $\frac{\partial(z,m)}{\partial m}$ can be interpreted as the Fokker-Planck equation for the SDE. The equivalence $\hat{\Psi} \equiv p^{(4)}$ holds up to an additive constant, which vanishes when applying the "$\nabla_{\log}$" operator or in the context of the Fokker-Planck equation (since all operators are linear). A similar interpretation applies to the PDE $\frac{\partial \Psi(z,m)}{\partial m}$ can be equivalently viewed from the reversed time coordinate as:

$$\begin{cases} \frac{\partial \Psi(z,s)}{\partial s} = \nabla \cdot (\hat{\Psi}f_s) + \frac{1}{2}\beta_s \Delta \Psi \\ \frac{\partial \hat{\Psi}(z,s)}{\partial s} = \nabla \Psi^T f_s - \frac{1}{2}\beta_s \Delta \hat{\Psi} \end{cases}, \tag{15}$$

where $s := 1 - m$. This implies that $\Psi(z, s)$ can be interpreted as the density (up to a constant factor) of the SDE as:

$$\mathrm{d}Z_m = f_m \mathrm{d}m + \sqrt{\beta_m}\mathrm{d}W_m, \quad Z_0 \sim \hat{\Psi}(\cdot, 0),$$
$$\mathrm{d}Z_m = f_m \mathrm{d}m + \sqrt{\beta_m}\mathrm{d}\overline{W}_m, \quad Z_1 \sim \Psi(\cdot, 1), \tag{16}$$

Eqn.(5) naturally follows by conditioning Nelson's duality [60], i.e., $q(\cdot, m) = \Psi(\cdot, m)\hat{\Psi}(\cdot, m)$, on a boundary pair $(Z_0, Z_1)$,

$$q(Z_m | Z_0, Z_1) = \Psi(Z_m, m | Z_0)\hat{\Psi}(Z_m, m | Z_1).$$

Because $\Psi(Z_m, m | Z_0)$ and $\hat{\Psi}(Z_m, m | Z_1)$ are solutions to the Fokker-Planck equations, we can express the posterior as the product of two Gaussian distributions:

$$\Psi(Z_m, m | Z_0)\hat{\Psi}(Z_m, m | Z_1)$$
$$= \exp(-\frac{1}{2}(\frac{||Z_m - Z_0||^2}{\sigma_m^2} + \frac{||Z_m - Z_1||^2}{\bar{\sigma}_m^2})) \tag{17}$$
$$= \mathcal{N}(Z_m; \frac{\bar{\sigma}_m^2}{\bar{\sigma}_m^2 + \sigma_m^2}Z_0 + \frac{\sigma_m^2}{\bar{\sigma}_m^2 + \sigma_m^2}Z_1, \frac{\sigma_m^2 \bar{\sigma}_m^2}{\bar{\sigma}_m^2 + \sigma_m^2} \cdot I),$$

where $\sigma_m^2 := \int_0^m \beta_m\, dm$ and $\bar{\sigma}_m^2 := \int_m^1 \beta_m\, dm$ represent the analytical marginal variances of the SDEs Eqn. 16 when $f := 0$. We demonstrate that $q(Z_m | Z_0, Z_m)$ is the marginal density of the DDPM posterior $p(Z_n | Z_0, Z_{n+1})$. First, observe that when $f := 0$, $p(Z_n | Z_0, Z_{n+1})$ takes the form of an analytic Gaussian:

$$p(Z_n | Z_0, Z_{n+1})$$
$$= \mathcal{N}(Z_n; \frac{\alpha_n^2}{\alpha_n^2 + \sigma_n^2}Z_0 + \frac{\sigma_n^2}{\alpha_n^2 + \sigma_n^2}Z_{n+1}, \frac{\sigma_n^2 \alpha_n^2}{\alpha_n^2 + \sigma_n^2} \cdot I), \tag{18}$$

where we define $\alpha_n^2 := \int_{m_n}^{m_{n+1}} \beta_m\, dm$ as the accumulated variance between two consecutive time steps $(m_n, m_{n+1})$. It is evident that at the boundary $m_n := m_{N-1}$, we obtain:

$$q(Z_{N-1} | Z_0, Z_N) = p(Z_{N-1} | Z_0, Z_N) \tag{19}$$

because $\alpha_{N-1} = \int_{m_{N-1}}^{m_N} \beta_m \, dm = \bar{\sigma}_{N-1}^2$. Assuming the relation holds at $m_{n+1}$, it is sufficient to demonstrate as shown in:

$$q(Z_n|Z_0, Z_N) \stackrel{?}{=} \int p(Z_n|Z_0, Z_{n+1})q(Z_{n+1}|Z_0, Z_N)dZ_{n+1}. \tag{20}$$

Since both $p$ and $q$ are Gaussians, the Gaussian with the mean as:

$$\frac{\alpha_n^2}{\alpha_n^2 + \sigma_n^2}Z_0 + \frac{\sigma_n^2}{\alpha_n^2 + \sigma_n^2}\left(\frac{\bar{\sigma}_{n+1}^2}{\bar{\sigma}_{n+1}^2 + \sigma_{n+1}^2}Z_0 + \frac{\sigma_{n+1}^2}{\bar{\sigma}_{n+1}^2 + \sigma_{n+1}^2}Z_N\right)$$
$$= \frac{\bar{\sigma}_n^2}{\bar{\sigma}_n^2 + \sigma_n^2}Z_0 + \frac{\sigma_n^2}{\bar{\sigma}_n^2 + \sigma_n^2}Z_N, \tag{21}$$

where we use the fact that $\bar{\sigma}_n^2 + \sigma_n^2$ is constant for all $n$ and that $\alpha_n^2 = \sigma_{n+1}^2 - \sigma_n^2 = \bar{\sigma}_n^2 - \bar{\sigma}_{n+1}^2$ by design. Similarly, the right-hand side of Eq.20 contains the covariance as:

$$\frac{\alpha_n^2\sigma_n^2}{\alpha_n^2 + \sigma_n^2} + \frac{\bar{\sigma}_{n+1}^2\sigma_{n+1}^2}{\bar{\sigma}_{n+1}^2 + \sigma_{n+1}^2}\left(\frac{\sigma_n^2}{\alpha_n^2 + \sigma_n^2}\right)^2$$
$$= \frac{\alpha_n^2\sigma_n^2(\bar{\sigma}_{n+1}^2 + \sigma_{n+1}^2) + \bar{\sigma}_{n+1}^2\sigma_n^4}{\sigma_{n+1}^2(\bar{\sigma}_{n+1}^2 + \sigma_{n+1}^2)} \tag{22}$$
$$= \frac{\sigma_n^2\left[\alpha_n^2(\bar{\sigma}_n^2 + \sigma_n^2) + (\bar{\sigma}_n^2 - \alpha_n^2)\sigma_n^2\right]}{\sigma_{n+1}^2(\bar{\sigma}_{n+1}^2 + \sigma_{n+1}^2)} = \frac{\sigma_n^2\bar{\sigma}_n^2}{\bar{\sigma}_n^2 + \sigma_n^2}.$$

We demonstrate the consistency of the continuous-domain bridge adaptation posterior with the DDPM backward posterior. This validates the continuous-domain bridge adaptation of knowledge transitions in our framework.

## B   Other Result

**Experimental results of order-I:** We provide detailed experimental results under order-I in Table 4. This setup reflects a standard incremental learning protocol, allowing for a fair comparison across methods. The results demonstrate the effectiveness of our approach under this specific task progression.

Table 4: The accuracy (%) of our method (Ours) on the MTIL benchmark with order-I. Each row shows the performance on each dataset for the model trained after the corresponding task. The metrics for Transfer, Average, Last, and Preserve are highlighted in color.

| | Aircraft | Caltech101 | CIFAR100 | DTD | EuroSAT | Flowers | Food | MNIST | OxfordPet | Cars | SUN397 | |
|---|---|---|---|---|---|---|---|---|---|---|---|---|
| Transfer | | 88.5 | 68.3 | 44.8 | 49.4 | 70.2 | 88.6 | 60.9 | 89.1 | 64.8 | 64.2 | 69.2 |
| Aircraft | 56.1 | 88.5 | 68.3 | 44.8 | 55.3 | 71.1 | 89.1 | 59.5 | 89.1 | 64.8 | 65.1 | |
| Caltech101 | 53.1 | 97.1 | 68.3 | 44.8 | 55.3 | 70.9 | 88.5 | 59.5 | 89.1 | 64.8 | 65.6 | |
| CIFAR100 | 53.3 | 95.4 | 89.4 | 44.8 | 44.1 | 70.8 | 88.5 | 59.5 | 89.1 | 64.8 | 65.6 | |
| DTD | 52.5 | 95.0 | 86.2 | 81.3 | 42.7 | 68.9 | 88.5 | 62.7 | 89.1 | 64.8 | 63.4 | |
| EuroSAT | 52.9 | 95.0 | 86.2 | 81.2 | 98.3 | 68.9 | 88.5 | 62.8 | 89.1 | 64.8 | 63.6 | |
| Flowers | 52.4 | 95.6 | 86.3 | 81.2 | 96.9 | 97.8 | 88.5 | 62.7 | 89.1 | 64.8 | 63.5 | |
| Food | 54.9 | 95.6 | 87.4 | 80.4 | 96.8 | 97.6 | 89.5 | 59.5 | 89.1 | 64.8 | 63.6 | |
| MNIST | 54.9 | 95.6 | 87.4 | 80.4 | 96.9 | 97.5 | 89.5 | 99.1 | 89.1 | 64.8 | 63.3 | |
| OxfordPet | 53.5 | 95.6 | 87.2 | 80.5 | 96.2 | 97.3 | 89.5 | 99.1 | 89.6 | 64.8 | 63.7 | |
| Cars | 53.6 | 95.6 | 87.6 | 80.3 | 97.5 | 97.3 | 89.5 | 99.1 | 89.6 | 88.8 | 63.8 | |
| SUN397 | 55.1 | 95.2 | 87.4 | 78.8 | 97.2 | 97.3 | 89.5 | 99.1 | 89.6 | 88.8 | 82.4 | 87.3 |
| Preserve | 53.7 | 95.4 | 87.0 | 80.4 | 96.9 | 97.4 | 89.5 | 99.1 | 89.6 | 88.8 | | 84.5 |
| Average | 53.9 | 94.9 | 83.8 | 70.8 | 79.8 | 85.1 | 89.1 | 74.8 | 89.3 | 69.2 | 65.8 | 77.9 |

**Mixup training:** Our LEBA adopts a phased training strategy: it first trains a new domain adapter independently, then fine-tunes it through cross-domain bridging, rather than mixing new and replayed data for joint optimization. This design choice is motivated by the significant distributional shift across domains—directly mixing replayed samples with current task data would require the model to

align with multiple domains simultaneously, which can hinder adaptation to the new task. As shown in the Table 5, our experimental results validate this observation. Moreover, since the amount of data for new tasks is typically much smaller than the replayed historical samples, joint training may lead to overfitting on past tasks and suppress the domain-specific adaptation required for the new task.

Table 5: Performance comparison between LEBA and mixup training strategy

|  | Transfer | Average | Last | Preserve |
|---|---|---|---|---|
| Mixup Training | 68.3 | 76.2 | 85.4 | 82.6 |
| Ours (LEBA) | 69.2 | 77.9 | 87.3 | 84.5 |

**Balance factor analysis:** We conduct ablation studies on the balance parameters $\gamma$ and $\beta$, as shown in the Tables 6 and 7. The best performance is achieved when $\gamma = 0.1$ and $\beta = 0.4$. Setting $\gamma$ too high (e.g., $\gamma > 0.1$) causes the CBA loss to dominate training, hindering the acquisition of new knowledge. Likewise, a large $\beta$ (e.g., $\beta > 0.4$) leads to overly smoothed integration weights, which diminish task-specific distinctions and negatively impact performance.

<table>
<tr><td colspan="5">Table 6: Sensitivity to balance factor $\gamma$</td><td colspan="5">Table 7: Sensitivity to balance factor $\beta$</td></tr>
<tr><td>$\gamma$</td><td>Avg</td><td>Last</td><td>Transfer</td><td>Preserve</td><td>$\beta$</td><td>Avg</td><td>Last</td><td>Transfer</td><td>Preserve</td></tr>
<tr><td>0.05</td><td>76.1</td><td>85.2</td><td>68.4</td><td>84.1</td><td>0.2</td><td>76.1</td><td>85.2</td><td>68.4</td><td>83.6</td></tr>
<tr><td>0.1</td><td>77.9</td><td>87.3</td><td>69.2</td><td>84.5</td><td>0.4</td><td>77.9</td><td>87.3</td><td>69.2</td><td>84.5</td></tr>
<tr><td>0.2</td><td>77.3</td><td>86.1</td><td>68.8</td><td>83.4</td><td>0.6</td><td>77.3</td><td>86.1</td><td>68.8</td><td>83.1</td></tr>
</table>

**Memory usage and training time:** We evaluated the training time and memory usage of our method compared to other methods. As shown in Table 8, the proposed LEBA framework achieves better computational efficiency compared to existing baselines. It requires less GPU memory and converges faster during training. This demonstrates that LEBA not only improves performance but also reduces resource overhead.

Table 8: Comparison of memory usage and training time

| Method | GPU (MiB) | Training Time (Min) |
|---|---|---|
| ZSCL [16] | 28,293 | 823.1 |
| MoE-Adapter [17] | 26,294 | 803.4 |
| Ours (LEBA) | 24,698 | 786.6 |

**Threshold $\eta$ analysis:** We investigate the effect of varying the threshold parameter $\eta$, which controls adapter selection in our method. As shown in Table 9, performance remains relatively stable across a range of $\eta$ values, indicating the robustness of our approach. Notably, the best overall performance is achieved when $\eta = 0.3$, suggesting an optimal balance between selective adapter reuse and new knowledge integration.

Table 9: Performance sensitivity to threshold $\eta$

| $\eta$ | Avg | Last | Transfer | Preserve |
|---|---|---|---|---|
| 0.1 | 76.6 | 85.9 | 68.3 | 84.6 |
| 0.3 | 77.9 | 87.3 | 69.2 | 84.5 |
| 0.5 | 77.5 | 86.8 | 68.8 | 83.7 |
| 0.7 | 77.4 | 86.6 | 68.5 | 83.6 |

**Other task order:** we randomized the task order and conducted two independent experiments based on the resulting orders. The corresponding results are presented in Table 9 and Table 9, with the task orders specified as: (a) [CIFAR100, DTD, Aircraft, Flowers, Food, StanfordCars, MNIST, EuroSAT, SUN397, OxfordPet, Caltech101] and (b) [EuroSAT, OxfordPet, SUN397, DTD, CIFAR100, Food, StanfordCars, MNIST, Caltech101, Flowers, Aircraft]. Experimental results demonstrate that the

proposed LEBA consistently achieves superior performance compared to state-of-the-art methods across different randomized task orders, indicating its robustness to task order variations.

Table 10: Comparison of methods across four evaluation metrics for task order (a)

| Method | Transfer | Average | Last | Preserve |
|---|---|---|---|---|
| ZSCL | 67.23 | 75.89 | 84.21 | 81.35 |
| MoE-Adapters | 68.67 | 76.21 | 85.36 | 82.65 |
| Ours(LEBA) | 69.42 | 77.66 | 87.13 | 84.01 |

Table 11: Comparison of methods across four evaluation metrics for task order (b)

| Method | Transfer | Average | Last | Preserve |
|---|---|---|---|---|
| ZSCL | 60.95 | 75.12 | 83.54 | 85.32 |
| MoE-Adapters | 61.57 | 75.26 | 84.92 | 86.16 |
| Ours(LEBA) | 62.46 | 76.69 | 86.62 | 88.93 |

