# OpenReview forum: "Learn and Ensemble Bridge Adapters for Multi-domain Task Incremental Learning"
_NeurIPS.cc/2025/Conference — NeurIPS 2025 poster_

### Official Review · Reviewer_SAmD · 2025-06-25

**Clarity:** 2
**Significance:** 3
**Originality:** 4
**Rating:** 4
**Confidence:** 4

**Summary:**

This paper introduces LEBA (Learn and Ensembling Bridge Adapters), a novel framework designed to tackle the challenging problem of Multi-domain Task Incremental Learning (MTIL). The core idea is to mitigate catastrophic forgetting and enhance knowledge transfer across a sequence of diverse domains. The framework is built upon two key contributions: (1) Continuous-domain Bridge Adaptation (CBA), which, for the first time, applies Schrödinger Bridge theory to model the knowledge transition between adapters as an optimal transport problem, aligning the feature distributions of new and past tasks; and (2) Progressive Knowledge Ensemble (PKE), a strategy that circumvents the need to store past data by using a diffusion-based generator for replay. PKE further enhances efficiency by maintaining a small, dynamic pool of useful historical adapters and using adaptive weights to create a more accurate representation of past knowledge. Through extensive experiments, the authors demonstrate that LEBA achieves state-of-the-art performance on the MTIL benchmark, outperforming existing methods.

**Questions:**

1. Could the authors elaborate on the trade-off between the flexibility of generative replay and the potential performance impact from distributional bias in the generated samples? How sensitive is the framework to the quality of the generator G?

2. The framework uses historical adapters' outputs as the target for alignment. How robust is this process if these historical adapters are themselves suboptimal? Do the PKE mechanisms (the dynamic pool and adaptive weights) sufficiently mitigate the risk of aligning to a flawed representation of the past?

3. While KL-divergence looks a reasonable choice, could the authors justify that this single metric is sufficient to capture the multi-faceted concept of an adapter's "usefulness"? Have alternative, more sophisticated selection mechanisms (e.g., based on diversity) been considered?

4. The fixed-size adapter pool ($P_K$) effectively ensures stability as the number of total tasks ($T$) grows. However, how does this stability mechanism interact with the replay batch size? Have the authors tested the framework's robustness with larger replay batches, and does a larger number of simultaneous alignment targets introduce new optimization challenges?

5. The authors state that the code will be released upon acceptance.  Could they confirm if the release will include the necessary assets and scripts to faithfully reproduce the main results presented in Tables 1 and 2?

**Ethical Concerns:**

["NO or VERY MINOR ethics concerns only"]

**Final Justification:**

Thanks a lot for the authors for their efforts to address the comments, which have been largely addressed. While I am not still fully convinced that the assumption (approximation) of the paper has been very well justified, I believe this revised version deserves a raised score.

**Limitations:**

Yes

**Quality:**

2

**Strengths And Weaknesses:**

**Strengths:**

1. The primary strength of this work lies in its novel formulation of the knowledge transfer problem in continual learning. Framing the alignment between past and present adapters as an optimal transport problem solvable by a Schrödinger Bridge is an interesting and original conceptual leap.  This approach moves beyond simple regularization or replay and introduces a more principled way to model the "pathway" of knowledge.

2. The LEBA framework is well-designed, addressing multiple facets of the MTIL problem. The synergy between the CBA and PKE modules is a key strength. While CBA provides the theoretical foundation for knowledge alignment, PKE offers a practical and flexible solution for revisiting past knowledge without the constraints of traditional replay methods (i.e., being independent of task order).

3. The paper provides compelling empirical evidence to support its claims. It compares LEBA against a strong and comprehensive set of baselines, including classic continual learning methods and recent state-of-the-art models, on a standard MTIL benchmark.  The use of multiple evaluation metrics and two different task orders (including a random one) demonstrates the robustness of the method.  Furthermore, the detailed ablation studies (e.g., on the adapter pool size and internal modules) effectively validate the contribution of each component of the proposed framework.

**Weaknesses:**

1. The framework relies on strong assumptions to make the elegant but complex theory computationally tractable. The Gaussian Bridge Path Assumption (Equation 5) is a significant simplification.  Real-world feature distributions from deep networks are often complex and multi-modal. Approximating the diffusion path as a simple Gaussian introduces a potential modeling error that could limit the optimality of the learned knowledge bridge.

2. The use of KL-divergence as a sole metric for adapter usefulness (Equation 8) is a heuristic simplification of a complex concept.  Usefulness could also encompass factors like knowledge diversity or complementarity, which may not be fully captured by a single similarity score.

3. The framework relies on samples from a pre-trained diffusion model.  These models are known to have their own biases and do not perfectly replicate the true data distribution, which could potentially mislead the alignment process.

4. The framework is reactive, evaluating the importance of a sample after it has been generated.  There is no mechanism to guide the generator G to proactively create samples that are predicted to be most beneficial for mitigating forgetting at a given training stage.

5. A key hyperparameter, the number of replayed samples  $|\hat{X}^t|$ per training step, is not explicitly mentioned, which hinders exact reproducibility.

---

> ### Author Rebuttal · Authors · 2025-07-31
>
> Thank you for your positive comments (novel formulation of the knowledge transfer problem,  well-designed). All your questions and weeknesses have been answered below.
>
> **Q1: Could the authors elaborate on the trade-off between the flexibility of generative replay and the potential performance impact from distributional bias in the generated samples? How sensitive is the framework to the quality of the generator?**
>
> **A1:** For the question you asked, it should refer to some noises when using generated samples, which might result into potential performance impact although generative replay is flexible. For this, we mitigate this risk from two below aspects: 1) the use of a high-quality pretrained diffusion model (DDPM) [1], which achieves strong fidelity and diversity (e.g., FID 3.17, IS 9.46 on CIFAR), and 2) our LEBA focuses on semantic-level alignment using adapter responses to high-level features, rather than relying on low-level pixel fidelity. This design choice enhances robustness and ensures that minor generation artifacts do not adversely impact the alignment or learning. To evaluate the sensitivity of our LEBA to the quality of the generator, we conducted experiments using different versions of the diffusion model, as shown in below table. This indicates that our LEBA exhibits low sensitivity to the choice of generator.
>
> Performance with different diffusion generator
> | Generator | Transfer | Average | Last  | Preserve |
> |-|-|-|-|-|
> | SD1.4| 69.2     | 77.9    | 87.3  | 84.5     |
> | SD1.5     | 69.03    | 77.73   | 87.48 | 84.31    |
> | SD2.1     | 68.94    | 78.12   | 87.51 | 84.74    |
>
> [1] Denoising Diffusion Probabilistic Models NIPS 2020.
>
> **Q2: If historical adapters are suboptimal, the PKE module mitigates this risk by adaptively weighting and integrating multiple prior adapters, reducing reliance on any single flawed representation and enhancing overall robustness.**
>
> **A2:** It is indeed common in incremental learning that historical adapters may be suboptimal, as no approach can fully preserve all prior knowledge perfectly over time. To address this, our framework incorporates the PKE module, which adaptively integrates $K$ historical adapters for each replay sample using learnable weights. This mechanism enables the model to prioritize more robust and relevant representations, rather than relying on any single adapter.
> In Table 3 of the manuscript, incorporating the PKE module which includes the use of these generated replay samples—results in a good performance gain across all evaluation metrics, demonstrating its effectiveness in mitigating the impact of imperfect historical adapters through selective and weighted knowledge integration.
>
> **Q3: While KL-divergence looks a reasonable choice, could the authors justify that this single metric is sufficient to capture the multi-faceted concept of an adapter's "usefulness"? Have alternative, more sophisticated selection mechanisms (e.g., based on diversity) been considered?**
>
> **A3:** We chose KL-divergence as our primary metric because it effectively captures the task-specific similarity between adapter outputs, aligning well with our goal of preserving semantically relevant knowledge.
> To further validate this choice, we also experimented with alternative metrics such as cosine distance and mutual information for adapter selection. As shown in the below table, these methods yield comparable performance across all four evaluation metrics, suggesting that KL-divergence is sufficiently effective in capturing the relevant aspects of an adapter’s “usefulness” in our setting.
>
> Comparison of adapter selection strategies on four evaluation metrics
> |Method|Transfer|Average|Last|Preserve|
> |-|-|-|-|-|
> |Cosine Similarity|68.94|77.61|86.84|84.86|
> |Mutual Information|69.37|77.73|87.17|84.37|
> |LEBA (Ours)|69.20|77.90|87.30|84.50|
>
> **Q4: The fixed-size adapter pool  effectively ensures stability as the number of total tasks grows. However, how does this stability mechanism interact with the replay batch size? Have the authors tested the framework's robustness with larger replay batches, and does a larger number of simultaneous alignment targets introduce new optimization challenges? A key hyperparameter, the number of replayed samples per-training step, is not explicitly mentioned, which hinders exact reproducibility.**
>
> **A4:**
> Firstly, we confirm that the fixed-size adapter pool provides stability as tasks accumulate, and we have evaluated the framework under varying replay batch sizes.
> To evaluate the interaction between this mechanism and replay batch size, we conducted experiments under different replay settings: 500$\times$16, 500$\times$32, and 500$\times$64 (i.e., generating 16, 32, or 64 replay samples per iteration over 500 iterations for each task). The results are shown in below table. As shown, performance gains saturate beyond the 500$\times$32 setting, indicating that LEBA is robust to variations in replay size. To balance performance and computational efficiency, we adopt the 500$\times$32 configuration as the default throughout our experiments.
>
> Secondly, in our LEBA, increasing the number of simultaneous alignment targets does not introduce notable optimization challenges, as the alignment operates at the semantic feature level with lightweight adapter outputs rather than raw high-dimensional data. As shown in below table, performance remains stable even when the number of replay samples increases, suggesting that the optimization process is not adversely affected.
>
> Performance on varying numbers of replayed samples
> |Replay number|Transfer|Average|Last|Preserve|
> |-|-|-|-|-|
> |500$\times$16|68.78|77.46|86.27|83.16|
> |500$\times$32 (Ours)|69.20|77.90|87.30|84.50|
> |500$\times$64|69.43|78.23|87.51|84.67|
>
> **Q5: The authors state that the code will be released upon acceptance. Could they confirm if the release will include the necessary  scripts to faithfully reproduce the main results?**
>
> **A5:** We commit to releasing the corresponding code and script resources that can reproduce the main results if the manuscript is accepted.
>
> **Q6: The framework relies on samples from a pre-trained diffusion model. These models are known to have their own biases and do not perfectly replicate the true data distribution, which could potentially mislead the alignment process.**
>
> **A6:** We would like to clarify that the replay samples used in the PKE module are generated by a pretrained diffusion model (DDPM) [1], which has demonstrated strong generative quality. For example, on the CIFAR dataset, DDPM achieves FID 3.17 and IS 9.46, indicating high fidelity and discriminability relative to the original data distribution.
> While generative models may introduce certain biases, our LEBA relies more on the semantic alignment between adapter representations than on pixel-level realism. The consistent performance gains observed in Table 3 of the manuscript suggest that the generated samples are sufficiently informative for stable knowledge alignment and replay.
> As part of future work, we plan to explore post-processing strategies, or employ more advanced generative models to further enhance the fidelity and diversity of replay data.
>
> [1] Denoising Diffusion Probabilistic Models NIPS 2020.
>
> **Q7: The framework lacks proactive guidance to generate samples optimal for mitigating forgetting.**
>
> **A7:** We have carefully reviewed the LEBA framework and concur that, as you noted, this aspect presents an opportunity for further enhancement. We appreciate your insightful suggestion and regard it as a direction for future improvements.

---

> ### Author Response · Authors · 2025-08-04
>
> Dear reviewer SAmD,
>
> Thank you sincerely for the time and effort dedicated to reviewing our manuscript and providing constructive feedback. We have detailed our responses to your queries and trust they adequately address your points. We remain fully available to discuss any further questions or clarifications you may have. We would greatly appreciate it if you could update your rating by synthesizing both the reviewers' comments and our corresponding replies.

---

> ### Comment · Area_Chair_thHi · 2025-08-06
> **Reminder**
>
> Dear Reviewer,
>
> This is a friendly reminder to check the authors' rebuttal and adjust your rating if necessary. Thanks for your contributions to the NeurIPS reviewing process.
>
> Thanks,
>
> Your AC

---

> > ### Comment · Reviewer_SAmD · 2025-08-06
> >
> > Thanks a lot for the authors for their efforts to address the comments, which have been largely addressed. While I am not still fully convinced that the assumption (approximation) of the paper has been very well justified, I believe this revised version deserves a raised score.

---

> ### Author Response · Authors · 2025-08-07
> **Thank you so much for acknowledging our rebuttal and supporting a score increase.**
>
> Dear reviewer SAmD,
>
> We sincerely appreciate your recognition of our rebuttal efforts and the time taken to re-evaluate our work. We are delighted to learn that most of your concerns have been addressed, and you are inclined to raise your score.
>
> Regarding the concern about the Gaussian Bridge Path assumption, while we acknowledge its approximate nature, it is theoretically grounded and has been widely adopted in generative model method [1,2].
>
>
> Thank you for your time and proposing these valuable comments/suggestions again!
>
> Authors
>
> [1] Diffusion schrödinger bridge matching NIPS 2023.
>
> [2] I$^2$SB: Image-to-Image Schrödinger Bridge ICML 2023.

---

### Official Review · Reviewer_VGXi · 2025-07-01

**Clarity:** 2
**Significance:** 3
**Originality:** 2
**Rating:** 4
**Confidence:** 4

**Summary:**

This paper presents a new framework for multi-domain task incremental learning (MTIL), called Learning and Ensembling Bridge Adapters (LEBA). LEBA contains two main technical components, continuous-domain bridge adaptation (CBA) and progressive knowledge ensemble (PKE). The CBA moduel aligns the latent distributions of current and previous adapters via Schrodinger Bridge, and the PKE module generates synthetic samples conditioned on previously seen semantic concepts (e.g., classes) using a pretrained diffusion model. Experiments consider two task orders and multiple benchmarks, showing that LEBA outperforms baselines such as ZSCL and MoE-Adapters.

**Questions:**

Please refer to the weaknesses above.

**Ethical Concerns:**

["NO or VERY MINOR ethics concerns only"]

**Final Justification:**

The rebuttal addresses several concerns and provides clarifications on task-order independence and sample fidelity, which are helpful. While some claims could still benefit from stronger empirical support, the authors’ willingness to temper statements and incorporate clarifications is encouraging. I will maintain a borderline rating, leaning toward acceptance.

**Limitations:**

Not clearly presented.

**Paper Formatting Concerns:**

No concerns

**Quality:**

3

**Strengths And Weaknesses:**

(Strengths)
1. The paper is well-organized and generally well-written, despite verbose descriptions in some theoretical parts.
2. Schrodinger bridge is firstly employed in the context of continual learning, which appeals some extent of novelty.
3. The proposed method achieves strong empirical results across multiple metrics, Transfer, Average, Last, and newly proposed Preserve.
4. The method is resource-efficient in memory and training time, making it suitable for practical MTIL applications.


(Weaknesses)
1. The claim that the method is “independent of task-learning sequence” is overstated. Replay samples are generated based on an accumulated set of prior class labels {$C_j$}$_{j=1}^{t-1}$, which are inherently tied to task order. Therefore, the replay process remains temporally dependent.
2. The use of semantic prompts alone (i.e., class labels) to condition the diffusion model assumes that these labels are sufficient to recover representative samples of previous domains. However, no evidence is provided to validate the quality or fidelity of these generated samples relative to the original data distributions.
3. The adapter pool size is fixed to K=2, and increasing it shows negligible gain, suggesting that either the replayed information is not diverse, or the adapter selection mechanism is too coarse.
4. The CBA module, while well-formulated, does not introduce novel theoretical contributions beyond adapting existing SB-based score matching to the adapter alignment setting.
5. The writing quality in the Related Work section could be improved for clarity and logical flow. For instance, the opening sentence refers vaguely to "the above method" without specifying which work is meant, creating confusion in a standalone section.
6. The presentation of Section 3.3 is overly technical and difficult to follow. The notation is heavy and inconsistent at times, and the core intuition (i.e., aligning latent distributions of adapters using Schrödinger Bridge dynamics( is obscured by excessive mathematical detail.

---

> ### Author Rebuttal · Authors · 2025-07-31
>
> Thanks for your positive comments on the well-organized structure, the extent of novelty, and the comprehensive experiments. The point-to-point answers are provided below.
>
> **Q1: The claim of being “independent of task order” is overstated, as replay relies on accumulated class labels, which are inherently tied to task sequence, making the process still temporally dependent.**
>
> **A1:** Thank you for raising this point. We would like to clarify our intended meaning behind the claim that our method is "independent of task-learning sequence". This statement refers specifically to the replay process in LEBA, which differs from standard replay-based methods that typically require explicit task identifiers or must associate each replayed sample with its original task or dataset. In contrast, LEBA generates replay samples based on an accumulated semantic set—i.e., the union of previously encountered class labels—without requiring knowledge of which specific task or dataset the sample originally belonged to.
> This task-agnostic design is the motivation for constructing a semantic concept set, and it enables flexible and unified replay across all previously learned knowledge, thus reducing the system’s reliance on task sequence during knowledge consolidation.
> We will revise this in the final version to better reflect this distinction and avoid overstating the claim.
>
> **Q2: Relying solely on class labels to condition the diffusion model assumes they suffice to recover representative samples, yet no evidence is provided to validate their fidelity to original distributions.**
>
> **A2:** Thank you for raising this point.
> We would like to clarify that the generated samples used in the PKE module are produced by a diffusion-based model (DDPM) [1], which has demonstrated strong generative quality—achieving FID 3.17 and IS 9.46 on CIFAR dataset, indicating sufficient fidelity and discriminability relative to the original data distribution.
> In the Table 3 of the manuscript, incorporating the PKE module which includes the use of these generated replay samples—results in a good performance gain across all evaluation metrics, confirming that the generated samples are informative and beneficial to the continual learning process.
> As part of future work, we plan to explore post-processing strategies, or employ more advanced generative models to further enhance the fidelity and diversity of replay data.
>
> [1] Denoising Diffusion Probabilistic Models NIPS 2020.
>
> **Q3: The adapter pool size is fixed and increasing it shows negligible gain, suggesting that either the replayed information is not diverse, or the adapter selection mechanism is too coarse.**
>
> **A3:** We would like to clarify that the negligible gain from increasing the adapter pool size $K$ is not attributed to a lack of diversity in the replayed information or a coarse adapter selection mechanism. First, our replay samples are generated via a diffusion-based generator conditioned on the accumulated semantic set, ensuring high semantic diversity across tasks without relying on raw data storage. Second, adapter selection is guided by KL divergence, which replaces only semantically redundant adapters and preserves a compact yet diverse pool.
> As illustrated in Figure 3 of the manuscript, the model performance tends to saturate when the adapter pool size reaches $K = 2$.
> As a result, expanding the adapter pool offers negligible performance gains.
> Specifically, LEBA employs learnable weights to assign importance scores to adapters based on their relevance to current replay samples. Consequently, larger values of $K$ lead to diminishing returns in both representational diversity and performance gains. Considering the minimal improvement beyond $K=2$ and the trade-off between memory overhead and model complexity, we adopt $K=2$ as the default setting to ensure efficient memory utilization.
>
> **Q4: The CBA module, while well-formulated, does not introduce novel theoretical contributions beyond adapting existing SB-based score matching to the adapter alignment setting.**
>
> **A4:** While the CBA module builds on established principles of Schrödinger Bridge (SB)-based score matching, its novelty lies in adapting and extending this framework to the unique setting of continual adapter alignment in multi-domain task incremental learning (MTIL).
> Unlike prior work that applies SB to generative modeling, we recontextualize it for cross-domain feature alignment across sequential tasks, using replayed samples and adapter responses as the distributions to be bridged.
> To our knowledge, this is the first work to apply SB-based methods for aligning modular representations in continual learning, offering a new perspective on stable knowledge transfer under task shifts.
>
> **Q5: The writing quality in the Related Work section could be improved for clarity and logical flow.**
>
> **A5:** Thank you for the comment. We will refine the Related Work in the next version to improve its writing quality.

---

> ### Author Response · Authors · 2025-08-06
> **Please let us know if any further questions! Thanks so lot!**
>
> Dear reviewer VGXi,
>
> Thanks for your timely reply! We would like to know **whether our responses have addressed your concerns**. We hope our rebuttal could address your concerns. We would appreciate it if you could **re-evaluate our submission** and we are looking forward to discussions if you have any other concerns. Thank you so much again!
>
> Authors

---

> > ### Comment · Reviewer_VGXi · 2025-08-07
> > **Post Rebuttal**
> >
> > This rebuttal resolves several concerns, but some claims remain only partially supported. In particular, the task-order independence and sample fidelity issues are addressed in spirit but not with strong empirical backing.
> >
> > I will maintain my borderline rating, leaning slightly toward acceptance, contingent on the paper clearly tempering some of its claims and incorporating the suggested clarifications in the final version.

---

> > > ### Author Response · Authors · 2025-08-07
> > > **Thank you so much for acknowledging our rebuttal.**
> > >
> > > Dear reviewer VGXi,
> > >
> > > We sincerely appreciate your thoughtful re-evaluation of our work and are pleased to know that your some concerns have been addressed through our rebuttal efforts.
> > > We will incorporate the suggested clarifications and appropriately temper some of our claims in the final version of the paper, in accordance with your feedback.
> > >
> > > For these questions you raised, we make the following clarification:
> > >
> > > **Q1: About task-order independence.**
> > >
> > > **A1:**
> > > Our method aims to construct a replay mechanism that does not depend on the task-learning sequence, which motivated our use of the phrase “independent of task-learning sequence.”
> > > We will revise this in the final version to avoid overstating the claim.
> > >
> > > **Q2: About sample fidelity issues.**
> > >
> > > **A2:**
> > > The PKE module utilizes samples generated by a diffusion-based model (DDPM) [1], which exhibits strong generative performance. Specifically, it achieves an FID of 3.17 and an IS of 9.46 on the CIFAR dataset, suggesting that the generated samples closely resemble the original data in terms of both fidelity and discriminative quality.
> > > We will incorporate this clarification into the final version in response to your suggestion.
> > >
> > > [1] Denoising Diffusion Probabilistic Models NIPS 2020.
> > >
> > > We sincerely appreciate the time and effort you dedicated to reviewing our work. We will carefully adopt your feedback into account in our future revisions. Once again, please accept our heartfelt thanks!

---

### Official Review · Reviewer_wSj1 · 2025-07-01

**Clarity:** 2
**Significance:** 2
**Originality:** 3
**Rating:** 4
**Confidence:** 3

**Summary:**

The paper addresses the multi-domain, task-incremental learning paradigm. The proposed method manages a pool of adapters from previous tasks. When training on a new task, the method aligns the feature distribution of the current adapter with those of previous ones. Instead of storing old data, the authors propose using a diffusion model to generate samples based on previously seen semantic concepts. The method is evaluated on two different orderings of 11 tasks.

**Questions:**

- What are the underlying reasons that a larger pool size (K) does not improve performance?
- How many replay samples do you generate, and how does this number affect performance?
- How is the "semantic set" actually computed? Is it based solely on class labels?
- In line 193, what metric is used to identify the "least representative" sample?
- Under what conditions do you replace an existing adapter, ignore the current one, or add a new one to the pool?
- How were the hyperparameters selected? Do you have recommendations for setting them on other benchmarks?
- Do you use the task ID at test time? Is the method applicable to multi-domain, class-incremental learning?
- How many random seeds were used for the experiments, and what is the standard deviation of the results?
- I expected that PKE combined with replaying generated samples would lead to higher gains than those reported in Table 3. Is there an intuition for why this module's contribution appears to be marginal?
- What are the exact benefits that PKE offers over standard replay for addressing task-order sensitivity?
- For heterogeneous learning, which approach is better: learning distinct knowledge specific to each domain, or forcing alignment between domains, which could be suboptimal if the domains are very different?

**Ethical Concerns:**

["NO or VERY MINOR ethics concerns only"]

**Final Justification:**

The authors addressed one of my main concerns regarding the task order and provided extra experiments for some unclear points.
There is still some limitations and points of improvements for future work. The authors provided a discussion for them in the rebuttal and would be valuable to be included in the revision.

**Limitations:**

Not thoroughly addressed.

**Paper Formatting Concerns:**

--

**Quality:**

3

**Strengths And Weaknesses:**

Strength:
- The paper addresses an interesting and important research problem that has received little attention: the presence of multiple, heterogeneous domains in a continual learning setting.
- It proposes a new way of replaying experiences using diffusion models.
- The proposed method achieves better performance than recent work.

Weakness:
- Some details are missing (see the questions section below).
- Some claims could be more strongly supported. For instance, the claim of robustness to task order could be better supported by studying more than two different orders.
- The computational cost of the approach and the usage of the diffusion model are not studied. (In line 309, the paragraph title is "Computational Cost," but it discusses memory costs instead.)
- It is unclear why an ensemble with more than two members does not help improve performance.

---

> ### Author Rebuttal · Authors · 2025-07-31
>
> Thanks for your positive comments (an interesting and important research,  a new way of replaying experiences). All your questions have been answered below.
>
> **Q1: Why does increasing the adapter pool size beyond two not yield further performance gains?**
>
> **A1:** To evaluate the effect of the adapter pool size, we conduct ablation studies varying $K$. As illustrated in Figure 3 of the manuscript, the model performance tends to saturate when the adapter pool size reaches $K = 2$. Consequently, adding more adapters introduces fluctuations.Our LEBA uses learnable weights to assign importance to each adapter based on its relevance to the current replayed sample. As a result, a larger $K$ provides diminishing returns in terms of both representational diversity and performance gains. Given the negligible performance improvement beyond $K=2$, and considering the trade-off between memory overhead and model complexity, we select $K=2$ as the default configuration to ensure efficient memory usage.
>
> **Q2: How many replay samples do you generate?**
>
> **A2:** We generate 32 replay samples per iteration over 500 iterations for each task. To further investigate the impact of replay quantity, we conduct additional experiments by generating training samples over 500 iterations with either 16 or 64 samples per iteration. The results are shown in the below table.To balance performance and computational cost, our LEBA adopts the 500$\times$32 replay setting, which achieves competitive results while maintaining efficiency.
>
> Performance on varying numbers of replayed samples
> |Replay number|Transfer|Average|Last|Preserve|
> |-|-|-|-|-|
> |500$\times$16|68.78|77.46|86.27|83.16|
> |500$\times$32 (Ours)|69.20|77.90|87.30|84.50|
> |500$\times$64|69.43|78.23|87.51|84.67|
>
> **Q3: How is the "semantic set" actually computed? Is it based solely on class labels?**
>
> **A3:** The semantic set refers to the collection of all classes encountered throughout the incremental learning process, and is defined solely based on the class labels.
> In the manuscript, we elaborate on the definition of the semantic set as " *The semantic set $\mathcal{C}^t \coloneqq \{c_j^t\}_{j=1}^{M_t}$ describes certain semantic information (e.g., class information $y_i^t$), with $M_t$ denoting the number of distinct class names.*"
>
> **Q4: In line 193, what metric is used to identify the "least representative" sample? Under what conditions do you replace an existing adapter?**
>
> **A4:** Please allow us to offer a further clarification. In line 193, the entity being evaluated as the "least representative" is not a training sample, but rather a task-specific adapter from the dynamic adapter pool. To determine whether a newly learned adapter should replace an existing one, we compute the KL-divergence between the current adapter and each adapter in the pool, based on their responses to the current task samples. A fixed threshold is employed to guide the replacement decision: if the divergence is below the threshold, the adapter pool remains unchanged; otherwise, the newly learned adapter replaces the most divergent one, thereby updating the pool accordingly.
>
> **Q5: How were the hyperparameters selected?**
>
> **A5:** We present the hyperparameter studies and corresponding rationale in the supplementary material (Lines \#519–\#523). The same hyperparameter settings are used for the other benchmarks.
>
> Sensitivity to balance factor $\gamma$
> |$\gamma$|Avg|Last|Transfer|Preserve|
> |-|-|-|-|-|
> |0.05|76.1|85.2|68.4|84.1|
> |0.1|77.9|87.3|69.2|84.5|
> |0.2|77.3|86.1|68.8|83.4|
>
> Sensitivity to balance factor $\beta$
> |$\beta$|Avg|Last|Transfer|Preserve|
> |-|-|-|-|-|
> |0.2|76.1|85.2|68.4|83.6|
> |0.4|77.9|87.3|69.2|84.5|
> |0.6|77.3|86.1|68.8|83.1|
>
> The best performance is achieved when $\gamma = 0.1$ and $\beta = 0.4$. Setting $\gamma$ too high (e.g., $\gamma > 0.1$) causes the CBA loss to dominate training, hindering the acquisition of new knowledge. Likewise, a large $\beta$ (e.g., $\beta > 0.4$) leads to overly smoothed integration weights, which diminish task-specific distinctions and negatively impact performance.
>
> **Q6: Do you use the task ID at test time? Is the method applicable to multi-domain class-incremental learning?**
>
> **A6:** Unlike class-incremental settings, task-incremental learning in MTIL typically assumes that the task ID is known at test time that follows the previous work [1,2]. Thus we also strictly follow the previous protocol in task-incremental learning [3]. Currently, our LEBA is not directly applicable to multi-domain class-incremental learning, as it is tailored for multi-domain task-incremental settings where task identities are known during training and inference, with each adapter aligned to a task-level distribution. In contrast, class-incremental learning requires unified classification without task labels, fundamentally altering the problem. Adapting our LEBA would require merging or aligning adapters across domains and classes, posing challenges in representation overlap and compatibility. We will explore such adaptations in future work.
>
> **Q7: How many random seeds were used for the experiments, and what is the standard deviation of the results?**
>
> **A7:** Following your suggestion, we have conducted experiments with five different random seeds and computed the standard deviations. The results are shown in the below table. The low standard deviations across all metrics demonstrate the robustness and stability of LEBA under randomized conditions.
>
> Performance and standard deviation of LEBA across four metrics
> |Metric|Transfer|Average|Last|Preserve|
> |-|-|-|-|-|
> |LEBA (Ours)|69.20|77.90|87.30|84.50|
> |Std. Dev.|±0.09|±0.23|±0.13|±0.28|
>
> **Q8: Why does PKE with generative replay yield only modest gains in Table 3?**
>
> **A8:** We would like to clarify that the PKE module actually includes the replay of generated data (as described in Line \#201). While the numerical improvements in Table3 may appear modest, the contribution of PKE lies not in delivering large standalone gains, but in enhancing robustness and consistency. Specifically, PKE adaptively integrates previously learned adapters during replay, facilitating stable knowledge reuse and sample-level alignment. These benefits are reflected in the consistent improvements observed across all four evaluation metrics.
>
> **Q9: What are the exact benefits that PKE offers over standard replay for addressing task-order sensitivity?**
>
> **A9:** Compared to standard data replay, the PKE module alleviates task-order sensitivity through two strategies.
> First, it introduces task-order independence by leveraging a dynamic adapter pool with learnable weights, enabling the adaptive integration of prior knowledge regardless of task sequence. Second, it supports semantic-level replay, where generated samples are conditioned on accumulated semantic concepts rather than stored per-task instances, resulting in more balanced and representative replay across tasks.
>
> **Q10: For heterogeneous learning, is it better to learn domain-specific knowledge or enforce cross-domain alignment?**
>
> **A10:** The issue you mentioned remains an open research challenge. Different methodological paradigms offer distinct advantages: learning domain-specific knowledge emphasizes specialization, while enforcing cross-domain alignment promotes generalization.
> Our work falls into the latter category—focusing on task alignment across domains via a shared adapter integration framework. Although this may introduce some suboptimality when domain gaps are large, we mitigate this risk through adaptive adapter selection and semantic-aware replay, which promote stable transfer while preserving task-relevant knowledge.
>
> **Q11: The robustness claim would be stronger with results on more than two task orders.**
>
> **A11:** According to your suggestion, we randomized the task order and conducted two independent experiments based on the resulting orders.
> The corresponding results are presented in blow table, with the task orders specified as: (a) [CIFAR100, DTD, Aircraft, Flowers, Food, StanfordCars, MNIST, EuroSAT, SUN397, OxfordPet, Caltech101] and (b) [EuroSAT, OxfordPet, SUN397, DTD, CIFAR100, Food, StanfordCars, MNIST, Caltech101, Flowers, Aircraft].
>
> Comparison of methods across four evaluation metrics for task order (a)
> |Method|Transfer|Average|Last|Preserve|
> |-|-|-|-|-|
> |ZSCL [1]|67.23|75.89|84.21|81.35|
> |MoE-Adapters [2]|68.67|76.21|85.36|82.65|
> |Ours (LEBA)|69.42|77.66|87.13|84.01|
>
> Comparison of methods across four evaluation metrics for task order (b)
> |Method|Transfer|Average|Last|Preserve|
> |-|-|-|-|-|
> |ZSCL[1]|60.95|75.12|83.54|85.32|
> |MoE-Adapters[2]|61.57|75.26|84.92|86.16|
> |Ours(LEBA)|62.46|76.69|86.62|88.93|
>
> Experimental results demonstrate that the proposed LEBA consistently achieves superior performance compared to state-of-the-art methods across different randomized task orders, indicating its robustness to task sequence variations.
>
> **Q12: In line 309, the paragraph title is "Computational Cost," but it discusses memory costs instead.**
>
> **A12:** Sorry for the typo Figure 4 of our manuscript. It should be corrected to "GPU Memory (MiB)". Figure 4 on the manuscript reports GPU memory consumption, and we will revise this error accordingly in the next version. Furthermore, Table 8 of the supplementary material presents a comparison of training time, as shown in the table below, where our LEBA demonstrates slightly better efficiency compared to current state-of-the-art methods.
>
> Comparison of memory usage and training time
> |Method|GPU (MiB)|Training Time (Min)|
> |-|-|-|
> |ZSCL|28293|823.1|
> |MoE-Adapter|26294|803.4|
> |Ours (LEBA)|24698|786.6|
>
> [1] Preventing zero-shot transfer degradation in continual learning of vision-language model ICCV 2023.
>
> [2] Boosting continual learning of vision-language models via mixture-of-experts adapters CVPR 2024.
>
> [3] Enhancing knowledge transfer for task incremental learning with data-free subnetwork NIPS 2023.

---

> > ### Comment · Reviewer_wSj1 · 2025-08-04
> >
> > I thank the authors for their response and running additional experiments. I think the new experiments on different task orders will support the paper claims.
> >
> > It is still unclear to me why there is no gains for k>2. In addition, I am not sure if "pool" would be the precise term to use in that case.
> >
> > Is the number of samples used in the replay-based baselines identical to that used in your proposed method?

---

> > > ### Author Response · Authors · 2025-08-05
> > > **Response to Reviewer wSj1 Comments**
> > >
> > > Thank you for your prompt response. We are pleased to hear that our previous reply addressed your concerns. We will now address your new questions one by one.
> > >
> > > **Q1: It is still unclear to me why there is no gains for k>2. In addition, I am not sure if "pool" would be the precise term to use in that case.**
> > >
> > > **A1:** point. To further clarify the performance gains for $K>2$, we conducted ablation studies by varying the adapter pool size $K$, as shown in Figure 3 of the manuscript.
> > > The results indicate a generally upward trend in performance as $K$ increases; however, the marginal gains diminish with larger pool sizes (e.g., $K$ = 6).
> > > We attribute this to the increased interference introduced during the replay process when too many historical adapters are retained.
> > > This phenomenon is not surprising and aligns with observations in Mixture-of-Experts (MoE) models, where an excessive number of experts can degrade performance [1].
> > > Therefore, we simply set a small $K$ in our LEBA to balance stability and effectiveness in the Progressive Knowledge Ensemble (PKE).
> > > The choice of $K$ remains an open research challenge, and future studies could explore adaptive mechanisms to determine its value more effectively.
> > >
> > > We use the term adapter pool to refer to a fixed-size set of task-specific adapters maintained throughout the continual learning process.
> > > It denotes a managed collection with a bounded capacity, where outdated or less relevant adapters are selectively replaced of our manuscript.
> > > We believe this terminology remains appropriate given the dynamic yet size-constrained nature of the set. We appreciate your suggestion and will consider clarifying this definition in the revised version.
> > >
> > > [1] Unchosen Experts Can Contribute Too: Unleashing MoE Models’ Power by Self-Contrast NIPS 2024.
> > >
> > > **Q2: Is the number of samples used in the replay-based baselines identical to that used in your proposed method?**
> > >
> > > **A2:** Thank you for your further question.
> > > While generative replay has been widely adopted in incremental learning settings (e.g., [1], [2], [3]), its integration into multi-domain task incremental learning (MTIL) remains largely unexplored.
> > > To ensure a further fair comparison, we extend the generative replay mechanism to the two most competitive baseline methods [4], [5] identified in the manuscript, under the same replay configuration as that adopted in the LEBA. That is, the number of generated samples per task is kept consistent across all methods. The corresponding results are shown in the table below.
> > > These results confirm that our LEBA maintains strong performance even under identical replay settings, highlighting its robustness and effectiveness.
> > >
> > > ### Table: Performance comparison of different methods on identical replay sample number
> > >
> > > | **Method**        | **Transfer** | **Average** | **Last** | **Preserve** |
> > > |-------------------|--------------|-------------|----------|--------------|
> > > | ZSCL [4]          | 68.34        | 75.76       | 84.15    | 81.67        |
> > > | MoE-Adapters [5]  | 68.96        | 76.98       | 86.47    | 83.13        |
> > > | Ours (LEBA)       | 69.20        | 77.90       | 87.30    | 84.50        |
> > >
> > > [1] DDGR: Continual Learning with Deep Diffusion-based Generative Replay ICML 2023.
> > >
> > > [2] SDDGR: Stable Diffusion-based Deep Generative Replay for Class Incremental Object Detection CVPR 2024.
> > >
> > > [3] Towards Efficient Replay in Federated Incremental Learning CVPR 2024.
> > >
> > > [4] Preventing zero-shot transfer degradation in continual learning of vision-language model ICCV 2023.
> > >
> > > [5] Boosting continual learning of vision-language models via mixture-of-experts adapters CVPR 2024.
> > >
> > > We sincerely hope that our respones have effectively addressed your concerns.

---

> > > > ### Comment · Reviewer_wSj1 · 2025-08-07
> > > >
> > > > Thank you for your response and running extra experiments.
> > > >
> > > > I believe my main concern of the paper regarding the task order is addressed during the rebuttal. For the rest of the comments, I strongly recommend including (extending) discussion on these points in the revision.
> > > >
> > > > I will update my score accordingly.

---

> > > > > ### Author Response · Authors · 2025-08-08
> > > > > **Thank you so much for acknowledging our rebuttal.**
> > > > >
> > > > > Dear reviewer wSj1,
> > > > >
> > > > > We sincerely appreciate your recognition of our rebuttal efforts and the time taken to re-evaluate our work. We are pleased to know that your concerns have been addressed and the rating updated accordingly.
> > > > >
> > > > > We make sure that the comments as suggested is incorporated into the revised manuscript, to further strengthen the validity and clarity of the paper.
> > > > >
> > > > >
> > > > > Your insights have been invaluable in enhancing the quality of our work.
> > > > >
> > > > > Authors

---

> ### Author Response · Authors · 2025-08-04
>
> Dear reviewer wSj1,
>
> We thank you sincerely for your time and effort in reviewing our manuscript and providing valuable suggestions. We have provided detailed responses to your questions and hope that they adequately address your concerns. If you need further clarification or have any other questions, please feel free to discuss them with us! We are more than willing to continue our communication with you. We would greatly appreciate it if you would update the rating by synthesizing other reviewers' comments as well as our responses.

---

### Official Review · Reviewer_zShr · 2025-07-03

**Clarity:** 4
**Significance:** 3
**Originality:** 3
**Rating:** 4
**Confidence:** 4

**Summary:**

The paper presents a framework for multi-domain task incremental learning that works by learning and using a cross-domain adapter to preserve previous knowledge as well as to transfer knowledge to new tasks. The contribution is two-fold: 1) a cross-domain adapter that aligns historical adapters through distribution transfer, and 2) a pool of adapters to store historically significant adapters. For 1), authors propose to use the Schrödinger Bridge (SB) mechanism to facilitate inter-task knowledge transfer by aligning probability distributions between current and previous adapters. For 2) the framework dynamically maintains an adapter pool of size K to store useful historical adapters. For each new adapter, its similarity to those in the pool is measured and the pool is updated accordingly. Experiments are run over a comprehensive set of datasets corresponding to 11 tasks, evaluating aspects like Transfer, Last task performance and Average performance. Additionally, forgetting is also measured. Ablation studies consider aspects such as removal of components and evaluation of the effect of hyperparameters.

**Questions:**

- What do you actually mean when you use the word "ensemble" in the context of the pool of adapters Pk?

**Ethical Concerns:**

["NO or VERY MINOR ethics concerns only"]

**Final Justification:**

Based on the discussion with authors, I will maintain my score.

**Limitations:**

Yes

**Paper Formatting Concerns:**

No formatting concerns

**Quality:**

3

**Strengths And Weaknesses:**

Strengths:
- The paper addresses an important problem of constructing cross-domain adapters in the multi-domain task incremental learning setting.
- The paper’s contributions seem novel in the sense that two different methods are proposed: one for performing cross-domain adaptation while learning tasks sequentially, and the other for selectively maintaining a pool of adapters that might me more useful for preserving knowledge as well as for future tasks.
- The paper is very well-written and presentation is excellent.

Weaknesses:
- The main weakness that I observe is in the experiments, where two possible task orders are used: alphabetically and random. Although I appreciate the consideration of multiple orders, I think these are insufficient to mitigate the effect that a particular task order may have on the results. I would expect to see at least a few (possibly random) orders, and average results of the metrics across these.
- Another weakness that I observe is in the description of the pool of adapters, where the word “ensemble” is repeatedly used. Based on the design proposed for this method as described in section 3.4, it does not entail an “ensemble” as known, but rather it’s merely a pool that gets updated depending on the similarity of new adapters with previous ones.

---

> ### Author Rebuttal · Authors · 2025-07-31
>
> Thank you for your positive feedback on the novelty and writing quality of LEBA. For your concerns, we make the response below.
>
> **Q1: The main weakness that I observe is in the experiments, where two possible task orders are used: alphabetically and random. Although I appreciate the consideration of multiple orders, I think these are insufficient to mitigate the effect that a particular task order may have on the results. I would expect to see at least a few (possibly random) orders, and average results of the metrics across these.**
>
> **A1:** In accordance with your suggestion, we randomized the task order and conducted two independent experiments based on the resulting orders.
> The corresponding results are presented in blow table, with the task orders specified as: (a) [CIFAR100, DTD, Aircraft, Flowers, Food, StanfordCars, MNIST, EuroSAT, SUN397, OxfordPet, Caltech101] and (b) [EuroSAT, OxfordPet, SUN397, DTD, CIFAR100, Food, StanfordCars, MNIST, Caltech101, Flowers, Aircraft].
> Experimental results demonstrate that the proposed LEBA consistently achieves superior performance compared to state-of-the-art methods across different randomized task orders, indicating its robustness to task sequence variations.
>
>
> Comparison of methods across four evaluation metrics for task order (a)
> |Method|Transfer|Average|Last|Preserve|
> |-|-|-|-|-|
> |ZSCL [1]|67.23|75.89|84.21|81.35|
> |MoE-Adapters [2]|68.67|76.21|85.36|82.65|
> |Ours (LEBA)|69.42|77.66|87.13|84.01|
>
> Comparison of methods across four evaluation metrics for task order (b)
> |Method|Transfer|Average|Last|Preserve|
> |-|-|-|-|-|
> |ZSCL[1]|60.95|75.12|83.54|85.32|
> |MoE-Adapters[2]|61.57|75.26|84.92|86.16|
> |Ours(LEBA)|62.46|76.69|86.62|88.93|
>
> Furthermore, by combining these results with those obtained under the randomized task order (Order II) presented in our manuscript, we compute the average performance across all four experimental settings, as shown below table.
> Experimental results demonstrate that our proposed LEBA consistently outperforms current state-of-the-art methods under randomized task orders.
>
> Average performance across three randomized task orders
> |Method|Transfer|Average|Last|Preserve|
> |-|-|-|-|-|
> |ZSCL[1]|64.09|75.17|83.71|82.72|
> |MoE-Adapters[2]|64.84|75.39|84.79|83.90|
> |Ours|65.66|75.65|86.71|86.24|
>
> [1] Preventing zero-shot transfer degradation in continual learning of vision-language model ICCV 2023.
>
> [2] Boosting continual learning of vision-language models via mixture-of-experts adapters CVPR 2024.
>
> **Q2: Another weakness that I observe is in the description of the pool of adapters, where the word “ensemble” is repeatedly used. Based on the design proposed for this method as described in section 3.4, it does not entail an “ensemble” as known, but rather it’s merely a pool that gets updated depending on the similarity of new adapters with previous ones.
> What do you actually mean when you use the word "ensemble" in the context of the pool of adapters Pk?**
>
> **A2:** We appreciate your comment and would like to provide additional clarification regarding the ensemble mechanism.
> Ensemble learning typically refers to the process of combining multiple models to improve robustness and generalization. In our LEBA, the ensembling process doesn't happen in the construction of the adapter pool itself, but through the adaptive integration of prior knowledge as defined in Eqn.(9) and Eqn.(10) of the manuscript.
> Specifically, we introduce a set of learnable parameters that assign sample-specific weights to each adapter in the pool. These weights are used to dynamically combine the outputs of historical adapters based on their relevance to the current task. This mechanism allows the model to selectively integrate previously learned knowledge in a principled and flexible manner, which aligns with the essence of ensemble learning.
> We will also make the distinction clearer between the adapter pool construction and the ensemble operation performed over it.

---

> > ### Comment · Reviewer_zShr · 2025-08-06
> > **Rebuttal read**
> >
> > Thanks to the authors for responding to my two concerns.
> >
> > Regarding the first one, just looking at the numbers, I observe a lot of variance in some of the performance metrics between the three task orders you have run (although you did not provide any measure of variance when you averaged them in your last table of the rebuttal!). Furthermore, are your results statistically significant across task orders, compared to the other methods? Again, I am not fully convinced of the effect of the order in the results, as some tasks may be more difficult for some methods, and this can have an important impact across the proposed method and other methods and how these compare to each other.
> >
> > I appreciate the answer to my second concern, and would suggest authors to clarify this further in the paper.
> >
> > I will keep my score unchanged.

---

> > > ### Author Response · Authors · 2025-08-07
> > > **Thank you so much for acknowledging our rebuttal.**
> > >
> > > **Q1: Regarding the first one, just looking at the numbers, I observe a lot of variance in some of the performance metrics between the three task orders you have run (although you did not provide any measure of variance when you averaged them in your last table of the rebuttal!). Furthermore, are your results statistically significant across task orders, compared to the other methods? Again, I am not fully convinced of the effect of the order in the results, as some tasks may be more difficult for some methods, and this can have an important impact across the proposed method and other methods and how these compare to each other.**
> > >
> > > **A1:**
> > > We sincerely appreciate your recognition of our rebuttal efforts. We are delighted to learn that some of your concerns have been addressed. For these questions you raised newly, we make the following clarification:
> > >
> > > i) We acknowledge that task order may influence the results, since some tasks might be more difficult for certain methods. However, a fully comprehensive evaluation is infeasible due to the combinatorial explosion of possible task permutations, for instance, 11 tasks (in our work) yield 11! (over 39 million). Thus, determining the optimal task order remains an open issue beyond the scope of our study. In our manuscript, to ensure fair comparisons, we strictly followed the previous protocols~[1,2] including the use of consistent task orders.
> > >
> > > ii) Following your suggestion, we compute the standard deviations across four metrics of experimental results, as presented in the table below. These methods exhibit a similar trend in terms of variance. At the same time, following your suggestion, we calculate the statistical significance (\textit{p}-values) of the average performance metric across the compared methods (see the table below), which reveal no significant differences.
> > > These results demonstrate that the comparison evaluation on methods is largely independent of task orders—provided that a consistent task order is used as input (which is also essential for fairness), even though a single method may have different performances on different task orders. Thus, the evaluation protocols used in [1,2] (also adopted in our work) are confident to evaluate continual learning alogrithms.
> > >
> > > **Table: Performance comparison (mean ± std) across different methods**
> > >
> > > | Method         | Transfer         | Average          | Last             | Preserve         |
> > > |----------------|------------------|------------------|------------------|------------------|
> > > | ZSCL [1]       | 64.09 ± 3.14     | 75.17 ± 0.70     | 83.71 ± 0.43     | 82.72 ± 2.27     |
> > > | MoE-Adapters [2] | 64.84 ± 3.58   | 75.39 ± 0.76     | 84.79 ± 0.64     | 83.90 ± 1.75     |
> > > | Ours (LEBA)    | 65.66 ± 3.51     | 76.65 ± 1.03     | 86.71 ± 0.37     | 86.24 ± 2.49     |
> > >
> > > **Table: _p_-values of the different methods**
> > >
> > > | Method         | ZSCL [1] | MoE-Adapter [2] | Ours (LEBA) |
> > > |----------------|----------|------------------|--------------|
> > > | _p_-Value      | 0.706    | 0.564            | 0.523        |
> > >
> > > [1] Preventing zero-shot transfer degradation in continual learning of vision-language model ICCV 2023.
> > >
> > > [2] Boosting continual learning of vision-language models via mixture-of-experts adapters CVPR 2024.
> > >
> > > Thanks for your feedback and suggestion again, and hope the above responses can solve your concerns.

---

> > > > ### Comment · Reviewer_zShr · 2025-08-08
> > > > **Answer**
> > > >
> > > > Thanks for the new table. As I hope you can see from this table and previous ones, there is a clear relationship between the task order and the performance (as would be expected not just for your method but for others as well). This leads to clear overlap between the performance of the proposed method and existing ones (except for the "Last" metric, which I hope you can see in the last table you reported). This is not necessarily bad, but that's precisely the reason why doing multiple runs with multiple task orders is important. I appreciate that you followed this advice.
> > > >
> > > > Based on this and other reviewers' comments and rebuttals, I will maintain my original score (which was already "borderline accept").

---

> > > > > ### Author Response · Authors · 2025-08-08
> > > > > **Thank you so much for acknowledging our rebuttal**
> > > > >
> > > > > Dear reviewer zShr,
> > > > >
> > > > > We sincerely thank the reviewer for the thoughtful feedback and for acknowledging our additional experiments and tables.
> > > > > Your constructive comments have been invaluable in refining our work, and we will carefully incorporate the suggested considerations into the revised manuscript.
> > > > >
> > > > > Thank you for your time and proposing these valuable comments/suggestions again!
> > > > >
> > > > > Authors

---

> ### Comment · Area_Chair_thHi · 2025-08-06
> **Reminder**
>
> Dear Reviewer,
>
> This is a friendly reminder to check the authors' rebuttal and adjust your rating if necessary. Thanks for your contributions to the NeurIPS reviewing process.
>
> Thanks,
>
> Your AC

---

### Note · Authors · 2025-08-13

Dear Reviewers, ACs, and PCs,

We sincerely appreciate your time and effort in evaluating our submission.
In this work, we propose the ``Learn and Ensemble Bridge Adapters (LEBA)" framework for multi-domain task-incremental learning. LEBA introduces key methodological innovations and is well validated through extensive experiments. We believe our work makes sufficient contributions to this topic of incremental learning and aligns well with the standards of NeurIPS.


The reviewers initially appreciated the novelty, originality, and well-written quality of this work.
During the rebuttal and discussion phase, we thoroughly their primary concerns, including task-order sensitivity, adapter-pool size, the quality and quantity of replayed samples, and additional experiments/analyses that further consolidate this work.
Following our responses, all reviewers offered positive follow-ups, notably, the two reviewers who had originally leaned toward "borderline reject" expressed a clear willingness to raise/update their ratings, while acknowledging the effectiveness of our approach, and the other two reviewers maintained their positive "borderline accept" scores.


We make sure to integrate the constructive suggestions raised during the rebuttal into the revised manuscript to further enhance clarity. We sincerely thank the reviewers for their thoughtful feedback and timely engagement.

Thanks for your attention.

Authors

---

### Decision · Program_Chairs · 2025-09-17

**Decision:**

Accept (poster)

**Comment:**

This paper works on the multi-domain task incremental learning problem. In the paper, the authors propose a continuous-domain bridge adaptation module, leveraging the distribution transfer capabilities of the Schrödinger bridge for stable progressive learning. It was reviewed by four expert reviewers, and all of them recommended accepting this paper after the rebuttal period. According to the reviewers, the paper is very well-written and the presentation is excellent. It also proposes a new way of replaying experiences using diffusion models. Therefore, it is a clear acceptance. The authors are encouraged to include the additional results and discussions during the rebuttal period in the final version.